# ChatGPT Based Data Augmentation for Improved Parameter-Efficient Debiasing of LLMs

**Pengrui Han**[1,2*], **Rafal Kocielnik**[1*], **Adhithya Saravanan**[1,3], **Roy Jiang**[1], **Or Sharir**[1]
**Anima Anandkumar**[1]
[1]California Institute of Technology, [2]Carleton College, [3] University of Cambridge
{barryhan@carleton.edu, rafalko@caltech.edu}

## Abstract

Large Language models (LLMs), while powerful, exhibit harmful social biases. Debiasing is often challenging due to computational costs, data constraints, and potential degradation of multi-task language capabilities. This work introduces a novel approach utilizing ChatGPT to generate synthetic training data, aiming to enhance the debiasing of LLMs. We propose two strategies: Targeted Prompting, which provides effective debiasing for known biases but necessitates prior specification of bias in question; and General Prompting, which, while slightly less effective, offers debiasing across various categories. We leverage resource-efficient LLM debiasing using adapter tuning and compare the effectiveness of our synthetic data to existing debiasing datasets. Our results reveal that: (1) ChatGPT can efficiently produce high-quality training data for debiasing other LLMs; (2) data produced via our approach surpasses existing datasets in debiasing performance while also preserving internal knowledge of a pre-trained LLM; and (3) synthetic data exhibits generalizability across categories, effectively mitigating various biases, including intersectional ones. These findings underscore the potential of synthetic data in advancing the fairness of LLMs with minimal retraining cost.

## 1 Introduction

Recent advancements in Large Language Models (LLMs) have significantly improved Natural Language Processing (NLP) capabilities. However, concerns about fairness have emerged (Bender et al., 2021b), highlighting that LLMs may inherit and amplify real-world biases like racial and gender biases (Kirk et al., 2021), toxicity (Gehman et al., 2020), and misinformation (Weidinger et al., 2022) due to their training on vast human-generated text data. This issue is critical when LLMs are applied in sensitive areas like healthcare, job recruitment, or criminal prediction, where such biases could lead to widespread discriminatory outcomes.

Considerable efforts have been made in recent research to debias LLMs. However, with the large size of these models, social bias mitigation appears to be particularly challenging (Xie & Lukasiewicz, 2023; Brown et al., 2020; Hoffmann et al., 2022). Traditional methods are computationally expensive as they often require model retraining (Tokpo et al., 2023). On top of that, retraining on limited data can lead to lowering LLM's general language capabilities due to catastrophic forgetting (Fatemi et al., 2023). On the contrary, recent parameter-efficient methods (He et al., 2022; Ding et al., 2022; Xie & Lukasiewicz, 2023) offer a good alternative as they only require minor and targeted parameter adjustments. While more efficient, these approaches heavily rely on the quality of training data (Delobelle et al.) and may offer a limited generalization, posing a challenge for comprehensive bias reduction (Li et al., 2022).

**Our Approach** In this work, to bolster the robustness of light-weight debiasing, we propose a method to systematically prompt ChatGPT (Ouyang et al., 2022) to generate

---

* Both authors contributed equally to this research

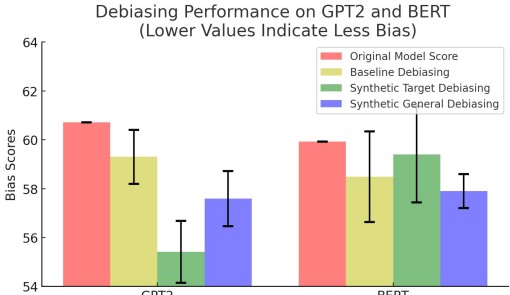

Figure 1: Debiasing performance of different strategies on GPT-2 and BERT averaged across three bias categories and two datsets (StereoSet and CrowS-Pairs).

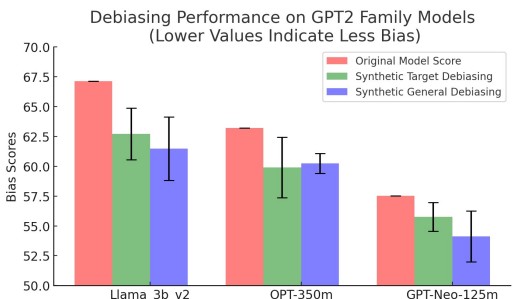

Figure 2: Average bias score across three bias categories and two metrics for different GPT2 family models before and after synthetic debiasing.

synthetic training data for LLM debiasing (Fig. 3). This is achieved using two distinct prompting strategies: targeted prompting and general prompting, complemented by an auxiliary method, loss-guided prompting. The first one is meant to debias models for a concrete category, which requires prior knowledge about the social bias to target, whereas the latter one has the potential to offer comprehensive debiasing and helps assess the generalizability of synthetic data to unknown social bias categories.

In particular, targeted prompting creates data specifically to address a particular category of bias, while general prompting generates data intended to be useful for mitigating bias across a range of categories. We conducted extensive evaluations on the impact of bias mitigation using our synthetic dataset through the parameter-efficient method of adapter tuning (Houlsby et al., 2019) across racial, gender, and religious bias. We also show promising results in debising models for challenging intersectional categories based on a recent BiasTestGPT dataset (Kocielnik et al., 2023b).

**Findings** Our primary findings include:

- Our synthetic data effectively mitigates bias in popular LLMs (Fig. 1). On GPT2 and BERT, our approach surpassed the performance of the recent Wikipedia-based dataset from Xie & Lukasiewicz (2023). Specifically, our best method enhances bias mitigation by an average of 6.4% on GPT-2 and 1.7% on BERT. Detailed results are in (Tables 2, 3, 5).
- Our method generalizes broadly reducing bias across GPT-2 family models by: 8.2% in LLaMA-3B, 5.8% in both OPT-350m and GPT-Neo-125m (Fig. 2).
- We also show promising results for challenging intersectional category related to Mexican Females from Kocielnik et al. (2023b) where we lower bias on GPT-2 by 12.9% (Table 6).
- As a result of our debiasing strategies, the general language model capabilities (LMS) in GPT-2 models are either slightly improved or minimally diminished (less than 1.3%). For BERT models, the variation in LMS is within 2.5%.
- Debiasing performance is improved with much less data, speeding up training by up to 60 times compared to Wikipedia-based baselines (Xie & Lukasiewicz, 2023).

**Contributions** We contribute by:

- Introducing a novel approach to bolster the robustness of parameter-efficient debiasing by prompting ChatGPT to generate high-quality synthetic debiasing data.
- Proposing two methods for synthetic data generation for debiasing: *targeted* - providing superior debiasing but requiring prior knowledge of social bias definition, and *general* - mitigating a range of social biases without prior knowledge but at the cost of reduced overall effectiveness.
- We further experiment with a variation of targeted prompting, a *loss-guided prompting*, that yields promising initial results on the BERT model.
- We share the code in our GitHub repository.

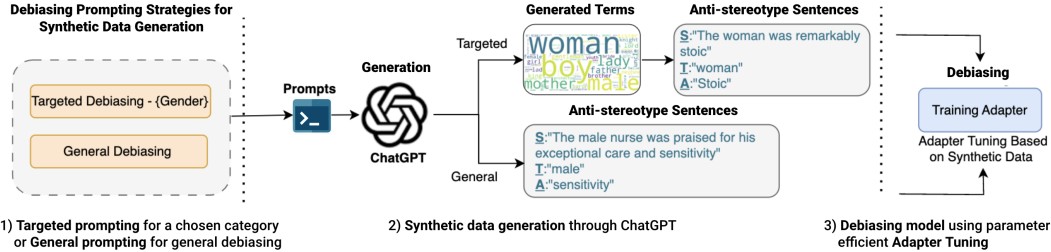

Figure 3: Our debiasing framework using ChatGPT-based synthetic dataset generation and Adapter-Tuning. The upper part is the process for targeted prompting and the bottom is for general prompting.

## 2 Related Work

### 2.1 Social Bias in Large Language Models (LLMs)

Biases can lead to harmful outcomes and may emerge at different stages of constructing and deploying an LLM (Suresh & Guttag, 2019). The primary causes include inherent biases in the training data and those introduced during the model construction stage, where the chosen architecture and algorithms may propagate certain biases (Ferrara, 2023; Bender et al., 2021a; Baumann et al., 2023).

Since LLMs like ChatGPT are trained on vast amounts of real-world data, limitations in current data preprocessing techniques can result in the retention of various biases within the pre-training dataset. These biases may include human stereotypes, underrepresentation of certain social groups, racism, hate speech, and other forms of bias (Ferrara, 2023; Zhao et al., 2019b; Bender et al., 2021a; He et al., 2023). Despite significant efforts to cleanse and curate datasets, the massive volume and complexity of the data often mean that some undesirable elements and imbalances in the representation persist. These data issues can potentially introduce biases to the language models during the training stage, and thus reproduce or even reinforce existing societal prejudices and inequalities (Li et al., 2023).

The model architecture and training algorithm themselves can introduce additional bias. Modern LLMs are designed to generalize from their training data to new, unseen inputs. This generalization is a double-edged sword. While it enables the models to understand text and sentences they have not previously encountered and to generate seemingly relevant responses to a wide range of queries, it also carries the risk of capturing only the central mass of the observed distribution, while failing to correctly model the tails. This can lead to perpetuating stereotypes and biased inferences (Ferrara, 2023). Bias mitigation in LLMs often involves a compromise in general performance and language capabilities. This trade-off represents a fundamental challenge in LLM bias mitigation efforts (Li et al., 2023).

### 2.2 Bias Mitigation Techniques

Studying and mitigating biases in LLMs has become increasingly important. Substantial effort has been devoted to addressing both algorithmic and training data bias. For algorithmic bias, prior work focused on developing model-agnostic human-understandable methods for detecting and mitigating biases (Richardson et al., 2023; DiCiccio et al., 2023). However, a significant challenge lies in the scarcity of suitable datasets, often relying on a limited number of benchmark datasets (Baumann et al., 2023). Additionally, these efforts do not extend to addressing biases inherent in the training data.

On the other hand, more general debiasing methods have been developed. These include novel algorithms (Yu et al.; Ma et al., 2020), leveraging pre-trained language models for generating gender variants of given texts (Jain et al., 2022), using unsupervised pipelines to curate and refine instances mentioning stereotypes (Gaci et al., 2023), increasing the amount of data used for training (Liang et al., 2020b; Schick et al., 2021a; Wang et al., 2022), and employing extra prompting to suppress social biases (Oba et al., 2023). However, these debiasing techniques often still rely on manual templates and are limited by the availability of the data (Delobelle et al.). Additionally, existing datasets have been shown to exhibit issues related to data quality and reliability (Blodgett et al., 2021). Furthermore, curated

datasets are often difficult to expand and generalize. Consequently, using them for debiasing could lead to overfitting specific social bias categories.

More importantly, even in the presence of proper data, the large-scale training methods that are not parameter-efficient or using post-hoc method (Liang et al., 2020a; Schick et al., 2021b) can be costly and potentially compromise the general language capabilities of an LLM (Xie & Lukasiewicz, 2023; Fatemi et al., 2023). This makes the process of debiasing a challenging endeavor. In this context, recent work of Xie & Lukasiewicz (2023) is noteworthy. They evaluated parameter-efficient debiasing methods on two popular language models, BERT (Devlin et al.) and GPT-2 (Radford et al., 2019), against gender, racial, and religious biases using existing datasets. Although the methods are more computationally efficient, they rely on a limited subset of the data from Wikipedia which can be used for counterfactual augmentation. The evaluation is also limited in scope to small LLMs. We contrast the effectiveness of the debiasing approach in this work to our own approach which utilizes dynamically generated synthetic data for bias mitigation. We further expand the debiasing to larger modern LLMs.

### 2.3 Synthetic Data Generation with Generative AI

As our debasing approach utilizes synthetic data generation, we provide a brief background of related methods in this area. Synthetic data plays a crucial role in scientific research. Commonly, it serves two primary purposes: 1) to emulate real-world data, enabling the creation of large datasets while preserving privacy (Angwin et al., 2022; Baumann et al., 2023), and 2) to generate test cases for evaluation purposes (Le Quy et al., 2022).

Traditionally, synthetic data generation has mainly relied on learning probability distributions from real data (Baumann et al., 2023). However, this approach often faces limitations, particularly when real data is scarce (Liu et al., 2024). Learning from limited data can also be costly and inflexible, requiring complex modeling techniques or human-designed frameworks (Guo & Chen, 2024; CRACS-INESC TEC & Faculty of Sciences, 2022). In contrast, generative AI, particularly LLMs trained on vast real-world data, can synthesize new data that closely resembles real data in both structure and content (Veselovsky et al., 2023). Previous studies have shown that LLMs can be used as effective training data generators for many applications (Borisov et al., 2022; Chintagunta et al., 2021; Sun et al., 2024). Recent research (Yu et al., 2024; Wang et al., 2023) further demonstrates that it is possible to control the quality and dimensions of the data during generation to achieve better outcomes. However, few studies have systematically developed strategies to generate efficient data specifically targeted for debiasing. In this work we leverage ChatGPT's knowledge of social bias to generate highly effective and generalizable data for mitigating bias in LLMs.

## 3 Methodology

Figure 3 introduces our debiasing framework and prompting strategies for synthetic data generation used for parameter-efficient LLM debiasing.

**Targeted Prompting:** In the targeted prompting approach, we prompt ChatGPT to produce sentences that aim to debias a specific category. Our first step is to identify the category of bias we aim to mitigate. The generation process consists of two primary components: term generation and sentence generation. Initially, we prompt ChatGPT to produce social group terms related to the chosen bias category by providing a few sample terms. Subsequently, we prompt ChatGPT to create anti-stereotyped sentences using the generated terms. We instruct ChatGPT to generate sentences that counter prevailing stereotypes associated with a particular social group (e.g., race-related terms). The desired output format is communicated by asking ChatGPT to produce sentences that connect a social group term with an anti-stereotyped attribute. Each generated sentence, "S", should also indicate the corresponding social group term, "T", and attribute term, "A", following the format: S, T, A. The previously generated terms serve as references for ChatGPT during this process.

To ensure the quality of the produced data, we include additional specific instructions. For instance, we ask ChatGPT to diversify the terms used and to produce sentences with varying

levels of complexity. All relevant terms can be found in Appendix 21 and 23. Examples of sentences and visualizations of terms are presented in Table 1 and Fig. 4 respectively. The prompts used for ChatGPT are detailed in Appendix 17.

**General Prompting:** The General Prompting approach aims to produce data that mitigates biases across various categories. Consequently, during the generation process, we afford ChatGPT greater freedom. We neither select specific bias categories nor generate social group terms. Instead, we directly prompt ChatGPT to create anti-stereotypical sentences that counteract stereotypes, adhering to the ["S","T","A"] format previously detailed. All terms are located in Appendix 22 and 24. Meanwhile, examples of sentences and visualization of terms are in Table 1 and Fig. 4. The ChatGPT prompts are in Appendix 17. *We formalize Targeted and General Prompting in Algorithm 1.*

**Loss-Guided Prompting:** We observed diminished effectiveness and a more pronounced trade-off between debiasing performance and language ability in models outside the GPT family, such as BERT, when using synthetic data generated from ChatGPT. This could be due to out-of-distribution generations from ChatGPT that harm the pre-trained knowledge of BERT in the course of further pre-training. A phenomenon known as catastrophic forgetting (Luo et al., 2023). To address this, we aim to guide ChatGPT to generate more in-distribution sentences for the given LLM. We select 50 samples exhibiting the highest and lowest loss, respectively under given LLM, from the generated data for each category. These samples, along with their corresponding loss scores, were then provided back to ChatGPT. This approach guides ChatGPT to generate data that is more in-distribution.

Loss-Guided Prompting is an auxiliary method used to generate more in-distribution data for targeted and general prompting, its format follows these two strategies, and we do not present it separately in the Table 1. *We formalize Loss-guided Prompting in Algorithm 2.*

**Training Methodology:** We train LLMs using synthetic data generated by GPT4 through adapter tuning (Houlsby et al., 2019). Adapter tuning freezes all the original parameters of an LLM and introduces, additional adapter layers into the original model architecture. Only these are trained for downstream applications. For GPT-2 and other GPT2 family models, we modify the sentence to position the attribute word at the end, employing the Causal Language Model loss as our training objective. For the BERT model, we mask the attribute word within the sentence and use the Masked Language Modeling (MLM) loss as the objective.

## 4 Experiment

**Metrics and Datasets:** To align with Xie & Lukasiewicz (2023), we use the CrowS-Pairs (Nangia et al., 2020) and the StereoSet (Nadeem et al., 2021) intrasentence datasets. The CrowS-Pairs dataset comprises pairs of contrasting sentences, one of which is more stereo-typed than the other. The StereoSet intrasentence dataset contains entries each composed of a stereotyped sentence, an anti-stereotyped sentence, and an unrelated sentence. The differences among these sentences are solely the attribute word. The CrowS-Pairs dataset contains 262, 105, and 516 entries for gender, religion, and race, respectively. For the StereoSet intrasentence set, there are 1026, 623, and 3996 examples respectively. For bias evaluations, we adopt the "stereotype score (SS)" from Meade et al. (2022). It quantifies the preference of an LLM for a stereotypical association over an anti-stereotypical one, with an ideal score being 50% for an unbiased model. To assess a model's general language capability, we use the "language modeling score (LMS)" from Nadeem et al. (2021). It measures the model's preference for meaningful associations over unrelated ones, with an ideal score of 100%.

**Training Details:** For the targeted prompting experiment, we generated three datasets for every category of bias, each containing 500 targeted sentences. For the general prompting experiment, we produced three datasets, each with 500 general sentences. We also tested various data sizes and selected 500 based on the optimal balance between debiasing performance and language ability impact. The performance graph for different data sizes is included in the Appendix 19. To represent masked and autoregressive language models, and to align with Xie & Lukasiewicz (2023), we chose to debias BERT (Devlin et al.) and GPT-2

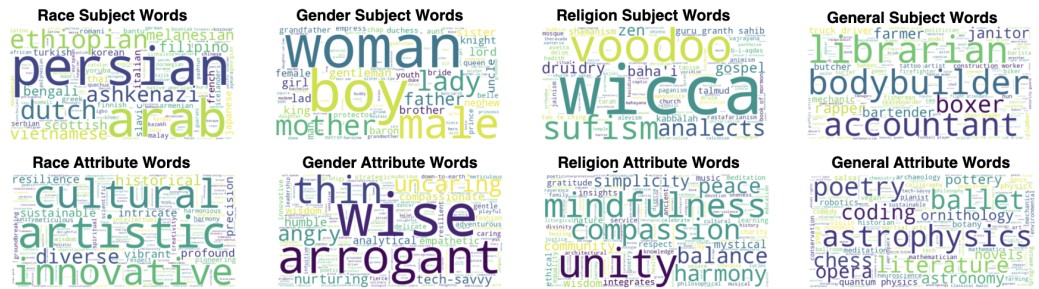

Figure 4: The most frequent words generated through each prompting are visualized via word clouds. The larger the word, the more frequently it has been generated.

| De-biasing Strategy | Bias Type | Sentence | Subject Word | Attribute Word |
|---|---|---|---|---|
| Targeted Prompting | Gender | "Love the Godfather not for his power, but for his genuine kindness." | "Godfather" | "Kindness" |
| | Race | "The Scottish community has been a beacon of innovation in various fields." | "Scottish" | "Innovation" |
| | Religion | "Many students find the Quran to be enlightening." | "Quran" | "Enlightening" |
| General Prompting | General | "Just because she's a cheerleader doesn't mean she isn't the top student in her calculus class." | "Cheerleader" | "Calculus" |
| | | "She found that the skateboarder was also a connoisseur of classical music." | "Skateboarder" | "Classical" |

Table 1: Example data of Targeted and General Prompting, including sentence, subject word, and attribute word for each example. Additional examples can be found in Appendix 16.

(Radford et al., 2019). To show the generalizability of our method, we also experimented with other GPT2 family models: Llama_3b_v2 (Touvron et al., 2023) (the latest version of LLaMA-3B model), OPT-350m (Zhang et al., 2022), and GPT-Neo-125m (Gao et al., 2020). We use Adapter Hub (Pfeiffer et al., 2020) and the code from Xie & Lukasiewicz (2023). We trained Llama_3b_v2 model on a Google Colab A100 GPU. All other experiments were conducted on a Google Colab V100 GPU. Based on empirical findings and the ratio between SS and LMS, we set the learning rate for the GPT-2 model to $5 \times 10^{-6}$. For BERT, the learning rate was set to $1 \times 10^{-5}$. For Llama_3b_v2 and OPT-350m, we used $5 \times 10^{-5}$, and for GPT-Neo-125m: $5 \times 10^{-4}$. For each bias category or for general debiasing, we conducted the experiments for the three datasets separately and reported the average outcomes as well as the standard deviations.

**Baseline:** For GPT-2 and BERT, we compare our debiasing approach, which uses synthetic datasets via adapter tuning, with other parameter-efficient methods and existing datasets, focusing particularly on the work of Xie & Lukasiewicz (2023). In their study, the authors down-sample 20% of the English Wikipedia as the debiasing corpus and augment it counterfactually for training (Zhao et al., 2019a; Zmigrod et al., 2019; Webster et al., 2020). The debiased corpus is then used with three distinct parameter-efficient methods: prefix tuning (Li & Liang, 2021), prompt tuning (Lester et al., 2021), and adapter tuning (Houlsby et al., 2019). For other models in the GPT-2 family, due to the lack of relevant prior work for comparison, we assessed the effectiveness of debiasing by comparing the debiased versions of the models to their original versions with weights before our debiasing.

## 5 Results

**Mitigating Racial Bias:** Table 2 shows that *our synthetic data surpasses all other parameter-efficient methods that utilize English Wikipedia for GPT-2 models* in mitigating racial bias. For BERT, our results are in line with the baselines. Our general debiasing achieves the best SS for StereoSet and yields results comparable to others for the SS on CrowS-Pairs, with the difference being less than 3%. In terms of language capability, our

| Racial Bias | CrowS-Pairs | Change↓ | StereoSet | Change ↓ | LMS | Change ↑ |
|---|---|---|---|---|---|---|
| **GPT-2 Model** | 59.69 | - | 58.90 | - | 91.01 | - |
| +Wiki-debiased + Prefix | $59.61_{0.51}$ | ↓0.13% | $57.53_{0.23}$ | ↓2.33% | $89.48_{0.08}$ | ↓1.68% |
| +Wiki-debiased + Prompt | $58.76_{0.92}$ | ↓1.56% | $57.72_{0.33}$ | ↓2.00% | $89.18_{0.10}$ | ↓2.01% |
| +Wiki-debiased + Adapter | $61.28_{1.27}$ | ↑2.66% | $57.77_{0.44}$ | ↓1.92% | $89.01_{0.68}$ | ↓2.20% |
| +Synthetic-targeted + Adapter * | $\underline{\mathbf{55.04}}_{3.63}$ | ↓7.79% | $\underline{\mathbf{47.35}}_{0.91}$ | ↓19.61% | $\mathbf{89.93}_{0.28}$ | ↓1.19% |
| +Synthetic-general + Adapter * | $58.79_{1.58}$ | ↓1.51% | $53.41_{0.96}$ | ↓9.32% | $88.74_{0.43}$ | ↓2.49% |
| **BERT Model** | 62.33 | - | 57.03 | - | 84.17 | - |
| +Wiki-debiased + Prefix | $57.44_{1.90}$ | ↓7.85% | $56.95_{0.39}$ | ↓0.14% | $\mathbf{84.35}_{0.12}$ | ↑0.21% |
| +Wiki-debiased + Prompt | $58.25_{3.90}$ | ↓6.55% | $58.17_{0.55}$ | ↑2.00% | $83.41_{0.80}$ | ↓0.90% |
| +Wiki-debiased + Adapter | $\mathbf{57.20}_{4.16}$ | ↓8.23% | $59.10_{0.45}$ | ↑3.63% | $84.34_{0.20}$ | ↑0.20% |
| +Synthetic-targeted + Adapter * | $61.75_{0.58}$ | ↓0.93% | $54.96_{2.23}$ | ↓3.63% | $81.48_{0.38}$ | ↓3.20% |
| +Loss-guided-targeted + Adapter * | $60.95_{0.64}$ | ↓2.21% | $55.02_{1.57}$ | ↓3.52% | $82.27_{0.59}$ | ↓2.26% |
| +Synthetic-general + Adapter * | $59.22_{0.89}$ | ↓4.99% | $\underline{\mathbf{54.84}}_{0.44}$ | ↓3.84% | $82.28_{0.17}$ | ↓2.25% |
| **LLaMA-3B Model** | 64.92 | - | 65.11 | - | 97.22 | - |
| +Synthetic-targeted + Adapter * | $61.43_{1.91}$ | ↓5.37% | $60.76_{1.02}$ | ↓6.68 % | $\mathbf{97.65}_{0.29}$ | ↑0.44% |
| +Synthetic-general + Adapter * | $\underline{\mathbf{60.21}}_{0.11}$ | ↓7.26% | $\mathbf{60.26}_{2.97}$ | ↓7.44% | $96.32_{0.64}$ | ↓0.93% |
| **OPT-350m Model** | 62.98 | - | 63.24 | - | 96.81 | - |
| +Synthetic-targeted + Adapter * | $\mathbf{58.91}_{1.96}$ | ↓6.46% | $\mathbf{56.26}_{2.39}$ | ↓11.04% | $\mathbf{97.37}_{0.16}$ | ↑0.58% |
| +Synthetic-general + Adapter * | $60.72_{0.91}$ | ↓3.59% | $60.43_{0.34}$ | ↓4.44% | $96.95_{0.01}$ | ↑0.14% |
| **GPT-Neo-125m Model** | 52.13 | - | 56.32 | - | 89.7 | - |
| +Synthetic-targeted + Adapter * | $51.74_{1.40}$ | ↓0.75% | $\mathbf{54.28}_{1.49}$ | ↓3.62% | $88.52_{0.66}$ | ↓1.32% |
| +Synthetic-general + Adapter * | $\underline{\mathbf{49.03}}_{1.36}$ | ↓5.95% | $54.35_{0.22}$ | ↓3.50% | $\mathbf{88.84}_{0.37}$ | ↓0.96% |

Table 2: Results on mitigating racial bias. "*" next to the method indicates our proposed approach. We present the average bias score with standard deviations from 3 runs paired with the % change compared to the model prior to debiasing. The first column lists the dataset and the parameter-efficient method employed. "Wiki-debiased" is baseline dataset from recent work (Xie & Lukasiewicz, 2023). "Synthetic-targeted" and "Synthetic-general" refer to our synthetic data generated via targeted and general prompting. "Prefix", "Prompt", and "Adapter" denote the three parameter-efficient methods. For instance, "+Synthetic-targeted + Adapter" means debiasing with synthetic data from targeted prompting using the adapter tune method. For both CrowS-Pairs and StereoSet datasets, a score closer to 50 (SS) is optimal, reflecting less bias. For the Language Model Score (LMS), a higher score is indicative of enhanced language capabilities. The positive direction of change is denoted in blue, while the negative is in red. The best score under each metric is marked in **bold** and underscored.

synthetic targeted approach secures the highest score on the GPT-2 model. For BERT, while our approach is outperformed by the baselines, the difference remains within 3.5%.

For the other models in the GPT-2 family, both targeted and general prompting strategies mitigate bias across both metrics, achieving an average reduction of 5.7% for targeted debiasing and 5.4% for general debiasing. Meanwhile, language ability is well-preserved: it is either slightly improved (approximately 0.5%) or minimally diminished (less than 1.3%).

**Mitigating Religious Bias:** As seen in Table 3, *our synthetic data outperforms all other methods in the baseline for the GPT-2 model.* While it slightly underperforms in LMS, the difference is marginal, at under 2%. For the *BERT model*, with the incorporation of loss-guided prompting, *our synthetic data achieves the best results* compared to all other methods in the baseline. In terms of LMS, the discrepancy is less than 2.5%.

For the other models in the GPT-2 family, general debiasing proves highly effective, yielding an average bias reduction of 7.2%. However, targeted debiasing is less effective, achieving an average reduction of only 0.7%. In terms of LMS, it is well preserved, exhibiting a variation of only 1.0% compared to the original LMS.

**Mitigating Gender Bias:** Our approach effectively reduces gender bias (Table 5). On GPT-2, our targeted data achieves the best SS on Stereoset, and our general data outperforms the baseline in the average SS score. In the case of BERT, although we did not surpass the baseline, with the implementation of loss-guided prompting, we still achieved an average bias reduction of 3.9% in loss-guided targeted debiasing and 2.6% in general debiasing. For the LMS, the difference is around 2.5%.

| Religious Bias | CrowS-Pairs | Change ↓ | StereoSet | Change ↓ | LMS | Change ↑ |
|---|---|---|---|---|---|---|
| **GPT-2 Model** | 62.86 | - | 63.26 | - | 91.01 | - |
| +Wiki-debiased + Prefix | $60.95_{0.60}$ | ↓3.03% | $65.16_{0.56}$ | ↑3.00% | $\mathbf{90.95_{0.03}}$ | ↓0.07% |
| +Wiki-debiased + Prompt | $58.29_{1.52}$ | ↓7.27% | $64.89_{1.52}$ | ↑2.57% | $90.68_{0.12}$ | ↓0.36% |
| +Wiki-debiased + Adapter | $62.10_{2.72}$ | ↓1.21% | $62.05_{0.66}$ | ↓1.92% | $90.31_{0.10}$ | ↓0.77% |
| +Synthetic-targeted + Adapter * | $\mathbf{57.78_{1.10}}$ | ↓8.09% | $\mathbf{59.72_{0.80}}$ | ↓5.58% | $89.35_{0.17}$ | ↓1.83% |
| +Synthetic-general + Adapter * | $\underline{58.73_{1.98}}$ | ↓6.55% | $62.44_{0.24}$ | ↓1.29% | $88.74_{0.43}$ | ↓2.49% |
| **BERT Model** | 62.86 | - | 59.77 | - | 84.17 | - |
| +Wiki-debiased + Prefix | $72.76_{1.55}$ | ↑15.76% | $60.61_{0.98}$ | ↑1.40% | $\mathbf{85.42_{0.09}}$ | ↑1.48% |
| +Wiki-debiased + Prompt | $83.05_{1.85}$ | ↑32.08% | $60.07_{1.12}$ | ↑0.50% | $83.80_{0.58}$ | ↓0.44% |
| +Wiki-debiased + Adapter | $68.00_{4.33}$ | ↑8.18% | $58.93_{1.19}$ | ↓1.40% | $84.45_{0.19}$ | ↑0.33% |
| +Synthetic-targeted + Adapter * | $62.86_{0.96}$ | ↓0.00% | $61.49_{5.35}$ | ↑2.87% | $82.48_{0.04}$ | ↓2.01% |
| +Loss-guided-targeted + Adapter * | $\mathbf{59.05_{1.14}}$ | ↓4.63% | $\mathbf{58.78_{2.93}}$ | ↓1.66% | $82.34_{0.24}$ | ↓2.17% |
| +Synthetic-general + Adapter * | $\underline{59.36_{1.29}}$ | ↓5.57% | $59.44_{0.75}$ | ↓0.55% | $82.28_{0.17}$ | ↓2.24% |
| **LLaMA-3B Model** | 75.24 | - | 63.69 | - | 97.22 | - |
| +Synthetic-targeted + Adapter * | $73.02_{1.98}$ | ↓2.95% | $61.64_{0.99}$ | ↓3.22% | $\mathbf{97.81_{0.32}}$ | ↑0.61% |
| +Synthetic-general + Adapter * | $\mathbf{63.81_{5.30}}$ | ↓15.19% | $\mathbf{60.52_{1.39}}$ | ↓4.98% | $96.32_{0.64}$ | ↓0.93% |
| **OPT-350M Model** | 59.05 | - | 64.62 | - | 96.81 | - |
| +Synthetic-targeted + Adapter * | $62.22_{1.98}$ | ↑5.37% | $63.80_{1.17}$ | ↓1.27% | $\mathbf{97.39_{0.26}}$ | ↑0.60% |
| +Synthetic-general + Adapter * | $\mathbf{57.78_{0.55}}$ | ↓2.15% | $\mathbf{62.15_{1.41}}$ | ↓3.82% | $96.95_{0.01}$ | ↑0.14% |
| **GPT-Neo-125M Model** | 55.24 | - | 62.72 | - | 89.7 | - |
| +Synthetic-targeted + Adapter * | $55.56_{1.45}$ | ↑0.57% | $60.97_{1.15}$ | ↓2.79% | $\mathbf{89.19_{0.31}}$ | ↓0.57% |
| +Synthetic-general + Adapter * | $\mathbf{48.89_{2.20}}$ | ↓11.5% | $\mathbf{59.18_{0.39}}$ | ↓5.64% | $88.84_{0.19}$ | ↓0.96% |

Table 3: Results on mitigating bias around Religion. "*" next to the method indicates our proposed approach. The terminologies and definitions follow those in Table 2.

Our method is also highly effective in reducing gender bias in other models in the GPT-2 family. We achieve an average reduction of 7.5% with targeted debiasing and 5.8% with general debiasing. The LMS varies within a margin of 1.0% compared to the original model.

## 5.1 General Conclusion for Results

**Debiasing is Effective:** Across all three categories of bias, our synthetic data, generated through both targeted, general, and loss-guided prompting, has demonstrated its effectiveness under different metrics. On GPT-2, our targeted debiasing approach reduced social bias by an average of 10.2% on StereoSet and 7.9% on CrowS-Pairs, while general debiasing achieved reductions of 5.3% and 5.1%. These figures surpass our baseline, which achieved average reductions of 2.5% on StereoSet and 2.2% on CrowS-Pairs. For BERT, our general debiasing approach reduced biases by 1.8% and 4.9%, exceeding existing methods with 1.6% and 3.2% reductions. However, targeted debiasing was less effective for BERT, showing no improvement on StereoSet and a 1.6% reduction on CrowS-Pairs. We addressed this by introducing a loss-guided targeted approach for BERT, enhancing results to 2.1% on StereoSet and 4.5% on CrowS-Pairs, thereby surpassing the baseline.

**Results Generalize Across LLMs:** We demonstrated broad generalizability in bias reduction across various GPT family models, including LLaMA-3B, OPT-350m, and GPT-Neo-125m, across three bias categories. For LLaMA-3B, bias was reduced by 7.1% and 6.8% using targeted and general strategies, respectively, on StereoSet, and by 6.2% and 9.7% on CrowS-Pairs. On OPT-350m, reductions were 5.3% and 4.6% on StereoSet, and 3.3% and 4.1% on CrowS-Pairs. GPT-Neo-125m showed decreases of 4.9% and 6.0% on StereoSet, and 1.0% and 5.7% on CrowS-Pairs.

**Targeted Prompting Usually More Effective:** Targeted prompting is more effective than general prompting in most cases. This is in line with our expectations that more prior knowledge leads to more robust debiasing. On the other hand, general debiasing compromises a bit of effectiveness in exchange for a broader range of bias mitigation.

**Debiasing & Language Capability Trade-off:** A noticeable trade-off emerges between language proficiency and bias mitigation when working with the BERT model. Although this trade-off was reduced through loss-guided prompting, it still presents an important focus of future exploration.

| Evaluator Model | Synthetic Data | Mean Score (SD) | Top 95% |
|---|---|---|---|
| ToxicBERT *(toxicity)* | General | $0.0084 \pm 0.0538$ | 0.0219 |
| | Gender | $0.0289 \pm 0.1125$ | 0.1302 |
| | Race | $0.0085 \pm 0.0370$ | 0.0314 |
| | Religion | $0.0115 \pm 0.0608$ | 0.0242 |
| VaderSentiment *(negative component)* | General | $0.0081 \pm 0.0404$ | 0.0000 |
| | Gender | $0.0318 \pm 0.0862$ | 0.2421 |
| | Race | $0.0039 \pm 0.0277$ | 0.0000 |
| | Religion | $0.0049 \pm 0.0350$ | 0.0000 |

Table 4: Toxicity and sentiment analysis of our synthetic data. The mean score represents the average toxicity or negative sentiment for all generated sentences across 3 runs in a given category with standard deviation (SD). The top 95% is the top percentile of the highest-scoring generations. The scores are from 0.0 to 1.0, where 1.0 is the highest toxicity or negative sentiment.

**Debiasing is Efficient:** Training costs—both in terms of time and memory—are substantially reduced. With smaller dataset than the baselines, we expedite the training process by approximately a factor of 60. We frequently secure results that match or surpass the baselines and original models in terms of bias mitigation and language ability.

# 6 Synthetic Dataset Analysis

**Dataset Similarity:** A natural concern arises that ChatGPT may know and merely reproduce the original test sets. We analyzed the similarity between the generated synthetic data and the test set. We compared the original StereoSet test set, the StereoSet development set, a different dataset, our synthetic dataset, and another StereoSet development set for various bias categories to check the uniqueness of our synthetic data. Table 7 in the Appendix reveals that for our synthetic dataset, the similarity matches that of a different dataset. For the targeted synthetic dataset, there is a pronounced similarity in terms of social group terms. This is anticipated because generating an extensive list of corresponding social group terms inevitably results in numerous overlaps and analogous terms. The authors of StereoSet manually ensured that the development and test sets did not share the same social group terms. We refrained from doing this to avoid referring to the test set.

**Unseen Biases:** To further ensure our synthetic data is not overfitting to the existing datasets, we use BiasTestGPT (Kocielnik et al., 2023b), which generates varied test sentences for different social categories and attributes through ChatGPT. While this dataset uses ChatGPT for sentence generation, the crucial social group and attribute terms defining bias categories are taken from psychology-backed studies from Guo & Caliskan (2021). We examine these biases for GPT-2 and BERT respectively (Table 6 in Appendix 14). For GPT-2, our debiasing effectively mitigates bias in a variety of categories including similar, intersectional, and less related categories. In the case of BERT, we observe a clear trade-off between language ability and bias mitigation, which aligns with our previous experiments.

**Sentiment & Toxicity Analysis:** We performed additional analysis of the potential toxicity and negative sentiment using unbiased ToxicBERT (Hanu & Unitary team, 2020) and VaderSentiment (Hutto & Gilbert, 2014). Table 4 shows that the scores for our synthetic data are very low, far below the 0.5 threshold for considering a text toxic (on a scale from 0.0 to 1.0). This is even for the top 95% percentile of highest-scoring generations. A manual inspection of the 55 sentences (out of all 6k generated) that scored above 0.5 on any of the dimensions revealed that toxicity detectors may be overly sensitive to certain keywords and contexts that, while potentially sensitive, are not inherently negative. Terms like "arrogant" or "angry," or references to specific nationalities or religious texts (e.g., "Serbian," "African," "the Analects of Confucianism"), seem to trigger automated flags, despite being used in a neutral or positive context. Similarly, unique combinations of roles and attributes meant to counter stereotypes seem misinterpreted (e.g., "garbage collector and a specialist in aero-dynamics"). Further details and scores for other dimensions of disruptive contents are presented in Table 9 in Appendix 20.

# 7 Discussion

We introduced synthetic data generation via targeted and general prompting to debias Large Language Models (LLMs). Our findings offer several avenues for deeper exploration.

**Efficacy of Prompting Strategies:** Our methodologies—targeted versus general prompting—vary in their approach and effectiveness across models. *Targeted Prompting* provides specificity in debiasing certain categories, while *General Prompting* offers a broader spectrum of bias mitigation. Notably, the effectiveness of these strategies demonstrated variation across models, such as GPT-2 and BERT, and different bias categories. One potential explanation is the difference in model architectures, affecting how each processes training data. Another reason could be the variance in training data, where different datasets or preprocessing methods influence the model's behavior. Finally, the specificity of bias categories might play a role, with targeted prompting being more effective for well-defined biases and general prompting for more complex or subtle biases. Further investigation is needed here.

**Understanding Trade-offs:** We observe a trade-off between language capability and bias mitigation, particularly pronounced in the BERT model (see graph in Appendix 15). This might be because the synthetic data is generated by ChatGPT, which significantly differs from BERT. We generate more in-distribution data through loss-guided prompting, which mitigates the issue, supporting the hypothesis. Nevertheless, the trade-off between the debiasing performance and the language ability is a fundamental problem (French, 1999). Understanding this trade-off becomes pivotal in different use cases and applications.

**Evaluating Synthetic Data's Universality:** Our similarity analysis underscores the uniqueness of our synthetic data, ensuring it isn't merely a reproduction of known datasets. Some robustness against different biases in another dataset - BiasTestGPT, suggests broader applicability. This is particularly relevant in an ever-evolving societal landscape with shifting norms and biases (Linegar et al., 2023; Kocielnik et al., 2023a). We note that further exploration of prompting strategies should investigate a good trade-off between prior knowledge of bias and the universality of the generated debiasing data.

# 8 Limitations

Our evaluation primarily relies on benchmarks and datasets with a North American English focus, which may not fully represent global biases. The effectiveness of our debiasing might vary in tasks outside our testing scenarios. There's also a concern that ChatGPT's exposure to test sets could have impacted our synthetic datasets (Prabhumoye et al., 2021). We investigated this possibility by checking if the synthetic data merely replicates known datasets and by experimenting with a newer dataset - BiasTestGPT. Nevertheless, alignment with test sets may still exist. Moreover, the dynamic nature of societal biases, which continually evolve, may require updates of our datasets. Our focus on explicit biases may overlook subtler ones (Goethals et al., 2024). This emphasizes the need for careful interpretation of our results and continuous improvement in debiasing approaches.

# 9 Conclusion

This paper presents two new methods for generating synthetic data to reduce social bias in LLMs more efficiently: general and targeted prompting. These methods outperform the recent work using parameter-efficient debiasing in bias mitigation and training efficiency. They also preserve language model capabilities. Our work highlights the potential of synthetic data in making LLMs fairer and suggests future research directions, including improving synthetic data generation, applying our approach to other domains such as vision, and exploring its broader applications beyond fairness.

# 10 Acknowledgements

We thank Caltech SURF program and Carleton's Wiebolt Endowed Internship Fund for contributing to the funding of this project. Anima Anandkumar is Bren Professor at Caltech. This material is based upon work supported by the National Science Foundation under Grant # 2030859 to the Computing Research Association for the CIFellows Project

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

## 11   Appendix - Gender Bias Mitigation Table

| Gender Bias | CrowS-Pairs | Change ↓ | StereoSet | Change ↓ | LMS | Change ↑ |
|---|---|---|---|---|---|---|
| **GPT-2 Model** | 56.87 | - | 62.65 | - | 91.01 | - |
| +Wiki-debiased + Prefix | $54.73_{0.66}$ | ↓3.76% | $61.35_{0.60}$ | ↓2.08% | $91.24_{0.07}$ | ↑0.25% |
| +Wiki-debiased + Prompt | $54.12_{1.14}$ | ↓4.84% | $61.30_{0.43}$ | ↓2.15% | $\mathbf{91.37}_{0.08}$ | ↑0.40% |
| +Wiki-debiased + Adapter | $\mathbf{52.29}_{1.13}$ | ↓8.05% | $60.33_{0.46}$ | ↓3.71% | $90.87_{0.11}$ | ↓0.15% |
| +Synthetic-targeted + Adapter * | $\underline{53.31}_{0.44}$ | ↓6.24% | $\mathbf{59.28}_{0.75}$ | ↓5.37% | $90.82_{0.39}$ | ↓0.21% |
| +Synthetic-general + Adapter * | $52.42_{1.17}$ | ↓7.79% | $59.77_{0.86}$ | ↓4.58% | $88.74_{0.43}$ | ↓2.49% |
| **BERT Model** | 57.25 | - | 60.28 | - | 84.17 | - |
| +Wiki-debiased + Prefix | $53.59_{0.19}$ | ↓6.39% | $57.82_{0.46}$ | ↓4.09% | $84.75_{0.15}$ | ↑0.69% |
| +Wiki-debiased + Prompt | $57.56_{1.41}$ | ↑0.54% | $58.07_{0.60}$ | ↓3.61% | $84.71_{0.16}$ | ↑0.64% |
| +Wiki-debiased + Adapter | $\mathbf{51.68}_{0.52}$ | ↓9.70% | $\mathbf{56.04}_{0.43}$ | ↓7.03% | $\mathbf{84.97}_{0.14}$ | ↑0.95% |
| +Synthetic-targeted + Adapter * | $54.96_{0.38}$ | ↓4.01% | $60.72_{0.50}$ | ↑0.73% | $79.20_{1.27}$ | ↓5.89% |
| +Loss-guided-targeted + Adapter * | $53.44_{0.44}$ | ↓6.66% | $59.55_{0.56}$ | ↓1.21% | $82.00_{1.68}$ | ↓2.58% |
| +Synthetic-general + Adapter * | $\underline{54.83}_{0.44}$ | ↓4.17% | $\underline{59.70}_{0.40}$ | ↓0.96% | $82.28_{0.17}$ | ↓2.24% |
| **LLaMA-3B Model** | 65.27 | - | 68.62 | - | 97.22 | - |
| +Synthetic-targeted + Adapter * | $\mathbf{58.52}_{3.82}$ | ↓10.34% | $\mathbf{60.88}_{3.29}$ | ↓11.27% | $\mathbf{97.27}_{0.38}$ | ↑0.05% |
| +Synthetic-general + Adapter * | $60.94_{4.52}$ | ↓6.63% | $63.22_{1.66}$ | ↓7.87% | $96.32_{0.64}$ | ↓0.93% |
| **OPT-350M Model** | 60.69 | - | 67.35 | - | 96.81 | - |
| +Synthetic-targeted + Adapter * | $\mathbf{55.34}_{1.15}$ | ↓8.82% | $\mathbf{62.90}_{6.61}$ | ↓3.6% | $\mathbf{97.37}_{0.08}$ | ↑0.58% |
| +Synthetic-general + Adapter * | $\underline{56.74}_{0.44}$ | ↓6.51% | $63.64_{1.38}$ | ↓5.51% | $96.95_{0.01}$ | ↑0.14% |
| **GPT-Neo-125M Model** | 54.96 | - | 63.74 | - | 89.7 | - |
| +Synthetic-targeted + Adapter * | $\mathbf{53.44}_{0.77}$ | ↓2.77% | $58.49_{0.98}$ | ↓8.24% | $\mathbf{89.18}_{0.04}$ | ↓0.60% |
| +Synthetic-general + Adapter * | $\underline{55.22}_{0.58}$ | ↑0.47% | $\mathbf{58.04}_{0.16}$ | ↓8.94% | $88.84_{0.19}$ | ↓0.96% |

Table 5: Results on mitigating gender bias. The terminologies and definitions follow those in Table 2.

## 12   Appendix - Targeted and General Prompting Algorithm

---

**Algorithm 1** Debiasing Data Generation for Targeted and General Prompting

---

**Input:** Bias category $N$ (optional for General Prompting), Generator Model $M_g$, Term generation instruction $i_t$, Targeted Prompting instruction $i_{tp}$, General Prompting instruction $i_{gp}$
**Output:** Debiasing Sentences $S$

1: **if** Targeted Prompting ($i_{tp}$) **then**
2:     $T \leftarrow \text{GENERATETERMS}(N, M_g, i_t)$
3:     $S \leftarrow \text{GENERATESENTENCES}(T, M_g, i_{tp})$
4: **else if** General Prompting ($i_{gp}$) **then**
5:     $S \leftarrow \text{GENERATESENTENCES}(M_g, i_{gp})$
6: **end if**
7: Reformat $S$: Sentence (S), Term (T), Attribute (A)
8: **return** $S$

---

## 13 Appendix - Loss Guided Prompting Algorithm

---

**Algorithm 2** Loss-Guided Debiasing Data Generation

---

**Input:** Debiasing sentences from Targeted Prompting $S_{tp}$, Generator Model $M_g$, Tested Model $M_t$, Loss-Guided Prompting instruction $i_{lgp}$, Number of loss-guided examples $k$
**Output:** In-Distribution Debiasing Sentences $S_{lp}$

1: $L_{tp} \leftarrow \{\}$
2: **for** $s \in S_{tp}$ **do**
3:      $l \leftarrow \text{EVALUATELOSS}(s, M_t)$
4:      Append tuple $(s, l)$ to $L_{tp}$
5: **end for**
6: $L_{tp} \leftarrow \text{SELECTHIGHLOWLOSS}(L_{tp}, k)$
7: $S_{lp} \leftarrow \text{GENERATESENTENCES}(L_{tp}, M_g, i_{lgp})$
8: Reformat $S_{lp}$: Sentence (S), Term (T), Attribute (A)
9: **return** $S_{lp}$

---

## 14 Appendix - Result Table for Testing on BiasTestGPT

| Model | Debiasing Category | Original SS | Gen. Debias | Tgt. Debias |
|---|---|---|---|---|
| GPT-2 | Profession <> Gender | 73.75 | $66.14_{3.09}$ | $\mathbf{64.46_{0.69}}$ |
| | Profession <> Math/Arts | **57.14** | $63.89_{1.04}$ | $64.18_{1.37}$ |
| | Mex.Fem<>Eur.Male/Emergent | 60.42 | $53.06_{0.64}$ | $\mathbf{52.64_{0.24}}$ |
| | Young <> Old | 55.94 | $52.92_{0.48}$ | $\mathbf{52.50_{0.62}}$ |
| BERT | Profession <> Gender | **66.76** | $68.88_{0.14}$ | $68.60_{0.28}$ |
| | Profession <> Math/Arts | 53.51 | $\mathbf{50.44_{0.44}}$ | $51.61_{0.50}$ |
| | Gender<>Science/Arts | **63.39** | $65.03_{0.68}$ | $64.44_{0.68}$ |
| | Gender<>Career/Family | **55.03** | $55.55_{1.01}$ | $55.13_{0.18}$ |

Table 6: De-biased Model Test Results Using BiasTestGPT Data. In this table, "<>" denotes bias between the chosen social categories. For instance, Profession <> Gender signifies bias between professional and gender terms. Mex.Fem<>Eur.Male/Emergent represents an intersectional category, indicating bias related to both race and gender. We employ synthetic data through general prompting for general de-biasing and synthetic gender data through targeted prompting for targeted de-biasing. The scores in the table are SS, with 50 being the ideal score.

## 15 Appendix - Dataset Comparison Table

| | Shared Terms (%) | | | Cos Similarity (%) | | |
|---|---|---|---|---|---|---|
| Dataset | Target | Attribute | Pairs | Sentence | Target | Attribute |
| **Gender** | | | | | | |
| StereoSet Gender Dev | 0.0 | 18.6 | 0.0 | 99.5 | 82.7 | 98.0 |
| CrowS-Pairs Gender | 76.7 | 23.3 | 4.5 | 96.1 | - | - |
| Synthetic-targeted* | $68.9_{1.6}$ | $10.5_{1.0}$ | $0.8_{0.4}$ | $96.1_{0.7}$ | $92.8_{0.1}$ | $85.2_{2.9}$ |
| Synthetic-general* | $8.9_{5.7}$ | $3.8_{0.2}$ | $0.1_{0.1}$ | $90.7_{0.6}$ | $\underline{56.4}_{4.2}$ | $71.9_{1.7}$ |
| StereoSet Religion Dev | 0.0 | 3.1% | 0.0 | 91.8 | 26.4 | 90.3 |
| **Race** | | | | | | |
| StereoSet Race Dev | 0.0 | 22.4 | 0.0 | 99.9 | 93.9 | 99.3 |
| CrowS-Pairs Race | 2.7 | 21.4 | 0.7 | 94.2 | - | - |
| Synthetic-targeted* | $\underline{19.5}_{0.0}$ | $4.5_{0.5}$ | $0.0_{0.0}$ | $94.1_{0.6}$ | $88.3_{0.2}$ | $89.2_{0.3}$ |
| Synthetic-general* | $\underline{2.4}_{2.7}$ | $3.9_{0.4}$ | $0.0_{0.0}$ | $93.1_{1.1}$ | $54.9_{3.0}$ | $84.9_{1.2}$ |
| StereoSet Religion Dev | 0.0 | 3.1 | 0.0 | 98.1 | 48.1 | 92.9 |
| **Religion** | | | | | | |
| StereoSet Religion Dev | 0.0 | 11.6 | 0.0 | 99.2 | 85.9 | 95.9 |
| CrowS-Pairs Religion | 22.2 | 17.4 | 0.9 | 93.9 | - | - |
| Synthetic-targeted* | $\underline{55.6}_{0.0}$ | $8.8_{0.7}$ | $0.6_{0.2}$ | $94.3_{0.4}$ | $87.9_{0.1}$ | $84.6_{1.3}$ |
| Synthetic-general* | $\underline{3.7}_{5.2}$ | $2.6_{0.7}$ | $0.0_{0.0}$ | $94.2_{0.7}$ | $\underline{40.5}_{0.5}$ | $72.0_{0.4}$ |
| StereoSet Profession Dev | 0.0 | 21.6 | 0.0 | 97.1 | 32.2 | 93.0 |

Table 7: Dataset Similarity Analysis. In this table, we compare both our synthetic data and other datasets with the test set. We provide the percentages of shared target terms, attribute terms, and shared pairs. Additionally, we evaluate the cosine similarities of the average word and sentence embeddings. The largest overlap between debising set and a test set is underlined.* denotes our generated dynthetic data.

## 16 Appendix - Generated Training Prompts Examples

| Debiasing Strategy | Bias Type | Sentence | Subject Word | Attribute Word |
|---|---|---|---|---|
| Targeted Prompting | Gender 1 | "The woman was remarkably stoic." | "Woman" | "Stoic" |
| | Gender 2 | "People love the Godfather not for his power, but for his genuine kindness." | "Godfather" | "Kindness" |
| | Gender 3 | "Contrary to belief, the Groomsman isn't careless; he's quite meticulous." | "Groomsman" | "Meticulous" |
| | Race 1 | "The Scottish community has been a beacon of innovation in various fields." | "Scottish" | "Innovation" |
| | Race 2 | "African architects are renowned for blending traditional and contemporary designs." | "African" | "Renowned" |
| | Race 3 | "In literature, Chinese philosophers have always explored the enigmatic nature of existence." | "Chinese" | "Enigmatic" |
| | Religion 1 | "Many students find the Quran to be enlightening." | "Quran" | "Enlightening" |
| | Religion 2 | "Many Torah scholars have contributed immensely to the fields of science." | "Torah" | "Science" |
| | Religion 3 | "Confucianism places great importance on family ties and respecting elders." | "Confucianism" | "family" |
| General Prompting | General 1 | "The male nurse was praised for his exceptional care and sensitivity." | "Male" | "Sensitivity" |
| | | "The football player wrote an award-winning poetry book." | "Football Player" | "Poetry" |
| | General 2 | "Her father took the day off to care for his child, showcasing his nurturing side." | "Father" | "Nurturing" |
| | | "Just because she's a cheerleader doesn't mean she isn't the top student in her calculus class." | "Cheerleader" | "Calculus" |
| | General 3 | "In many communities, it's the men who are the primary gossipers." | "Men" | "Gossipers" |
| | | "She found that the skateboarder was also a connoisseur of classical music." | "Skateboarder" | "Classical" |

Table 8: This table showcases example prompts. For Targeted Prompting, we provide an example for each generation of every category. For General Prompting, we provide two examples for each generation. Each example includes the sentence, subject word, and attribute word.

## 17 Appendix - ChatGPT prompts

**Prompts for Targeted Term Generation:** The following link is the conversation with ChatGPT we used for targeted terms generation:
`https://chat.openai.com/share/214c9ff0-dfc1-4111-b5c4-bb896ebd0c9b`

**Prompts for Targeted Sentence Generation:** We include the sample conversations with ChatGPT for Targeted Sentence Generation listed below:

1. Sample Conversation for Racial Bias:
   `https://chat.openai.com/share/252a3c4d-2295-45bd-b27d-75a277829d6a`

2. Sample Conversation for Gender Bias:
   `https://chat.openai.com/share/7ec33baa-e2e0-44dd-bb78-cbe63def1f80`

3. Sample Conversation for Religious Bias:
   `https://chat.openai.com/share/8ee8285d-c169-456a-a4fe-e48e8399c34b`

**Prompts for General Sentence Generation:** The following link is a sample conversation with ChatGPT we used for generating general de-biasing sentences:
`https://chat.openai.com/share/00dbd00c-fb14-4800-b699-9235093e716d`

## 18 Appendix - Trade-off Graph

Figure 5: This graph illustrates a clear trade-off between the model's language capabilities and debiasing performance during training. Lowering bias in a language model is likely to impact its general language proficiency. This represents a fundamental challenge in the field of language model fairness.

## 19 Appendix - Performance Graph for Different Data Sizes

Figure 6: Performance across different data sizes. The 200 data size yields minimal debiasing performance, while the 1000 data size significantly impairs the model's language capability. Thus, to achieve a balance between debiasing performance and language capability, a data size of 500 is selected.

## 20 Appendix - Additional Details of Toxicity and Sentiment Analysis

VaderSentiment (Hutto & Gilbert, 2014) produces individual positive, negative, and neutral sentiment component scores and a compound score. We report just the values for the negative sentiment component as these are the most relevant here. Aside from general Toxicity presented in Table 4, we also analyzed all the dimensions provided by ToxicBert (Hanu & Unitary team, 2020) - Severe Toxicity, Identity Attack, Insult, Sexually Explicit, Obscene, and Threat. We present the results for these other dimensions in Table 9.

| Evaluated Dimension | Synthetic Data | Mean Score (SD) | Top 95% |
|---|---|---|---|
| Toxicity | General | $0.0084 \pm 0.0538$ | 0.0219 |
| | Gender | $0.0289 \pm 0.1125$ | 0.1302 |
| | Race | $0.0085 \pm 0.0370$ | 0.0314 |
| | Religion | $0.0125 \pm 0.0608$ | 0.0242 |
| Obscene | General | $0.0002 \pm 0.0010$ | 0.0005 |
| | Gender | $0.0010 \pm 0.0071$ | 0.0020 |
| | Race | $0.0002 \pm 0.0007$ | 0.0027 |
| | Religion | $0.0006 \pm 0.0045$ | 0.0006 |
| Identity attack | General | $0.0012 \pm 0.0124$ | 0.0015 |
| | Gender | $0.0010 \pm 0.0049$ | 0.0028 |
| | Race | $0.0056 \pm 0.0350$ | 0.0151 |
| | Religion | $0.0023 \pm 0.0087$ | 0.0108 |
| Insult | General | $0.0053 \pm 0.0498$ | 0.0066 |
| | Gender | $0.0235 \pm 0.1105$ | 0.0855 |
| | Race | $0.0018 \pm 0.0105$ | 0.0049 |
| | Religion | $0.0011 \pm 0.0060$ | 0.0031 |
| Threat | General | $0.0001 \pm 0.0005$ | 0.0002 |
| | Gender | $0.0001 \pm 0.0006$ | 0.0003 |
| | Race | $0.0001 \pm 0.0002$ | 0.0002 |
| | Religion | $0.0001 \pm 0.0001$ | 0.0003 |
| Sexually Explicit | General | $0.0001 \pm 0.0004$ | 0.0003 |
| | Gender | $0.0002 \pm 0.0006$ | 0.0009 |
| | Race | $0.0003 \pm 0.0042$ | 0.0002 |
| | Religion | $0.0090 \pm 0.0764$ | 0.0003 |

Table 9: Analysis using additional dimensions of offensive content provided by ToxicBert (Hanu & Unitary team, 2020).

# 21 Appendix - Subject words for Targeted Prompting Data

Table 10: Subject Words for Gender Bias Data through Targeted Prompting

| Gender Generation 1 | | | | | |
|---|---|---|---|---|---|
| woman(36) | male(27) | boy(26) | girl(21) | female(20) | father(19) |
| man(18) | mother(18) | sister(17) | brother(17) | grandfather(15) | grandmother(11) |
| lady(12) | gentleman(11) | wife(9) | son(7) | uncle(6) | lord(5) |
| empress(5) | daughter(7) | mister(4) | sir(5) | mrs.(4) | miss(4) |
| patriarch(4) | knight(4) | baron(4) | queen(6) | madame(4) | king(7) |
| prince(5) | actress(3) | husband(5) | young lady(3) | guy(6) | lad(6) |
| emperor(3) | dame(3) | nephew(4) | duke(3) | bride(4) | maiden(3) |
| matron(3) | son-in-law(3) | mom(3) | dad(4) | gal(4) | mr.(3) |
| duchess(3) | businesswoman(2) | businessman(2) | granddaughter(2) | sister-in-law(3) | lass(2) |
| aunt(3) | matriarch(2) | maid(2) | grandson(2) | papa(2) | niece(3) |
| missus(2) | madam(1) | mum(1) | gent(1) | young man(1) | groom(2) |
| brother-in-law(2) | soldier(1) | ms.(1) | masculine(1) | boyfriend(1) | daughter-in-law(1) |
| count(1) | chap(1) | youth(1) | sire(1) | heir(1) | junior(1) |
| mother-in-law(1) | she(1) | princess(1) | heroine(1) | hostess(1) | bachelorette(1) |
| belle(1) | mummy(1) | bridesmaid(1) | mama(1) | bestie(1) | hero(1) |
| vixen(1) | goddess(1) | squire(1) | damsel(1) | bachelor(1) | countess(1) |
| maternal(1) | elder(1) | groomsman(1) | host(1) | heiress(1) | protector(1) |
| buddy(1) | baroness(1) | godfather(1) | ma(1) | | |
| Gender Generation 2 | | | | | |
| youth(12) | lord(12) | knight(12) | king(10) | uncle(10) | lad(10) |
| duchess(9) | baron(9) | bride(9) | nephew(9) | protector(9) | belle(8) |
| chap(8) | lady(8) | gentleman(8) | aunt(7) | countess(7) | groom(7) |
| empress(7) | mother(7) | prince(7) | mister(7) | godfather(6) | sir(6) |
| heroine(6) | duke(6) | boy(6) | queen(6) | maid(6) | sire(6) |
| buddy(6) | maternal(6) | bachelorette(6) | maiden(5) | groomsman(5) | son(5) |
| gal(5) | heir(5) | patriarch(5) | missus(5) | bachelor(5) | matron(5) |
| damsel(5) | count(5) | princess(5) | hero(5) | junior(5) | mummy(5) |
| best man(5) | daughter(5) | niece(5) | sister-in-law(5) | dame(5) | hostess(5) |
| son-in-law(4) | madame(4) | mother-in-law(4) | bridesmaid(4) | squire(4) | stag(4) |
| vixen(4) | daughter-in-law(4) | baroness(4) | lass(4) | male(4) | host(4) |
| matriarch(4) | father(4) | brother-in-law(3) | Mr.(3) | master(3) | Miss(3) |
| elder(3) | girlfriend(3) | boyfriend(3) | bestie(3) | wife(3) | sister(3) |
| man(3) | brother(3) | goddess(3) | motherhood(3) | grandson(3) | girl(3) |
| woman(3) | mademoiselle(2) | mom(2) | Pa(2) | granddaughter(2) | husband(2) |
| madam(2) | grandfather(2) | grandmother(2) | godmother(2) | mistress(1) | Ma(1) |
| Mama(1) | dad(1) | female(1) | Mrs.(1) | father-in-law(1) | feminine(1) |
| guy(1) | papa(1) | he(1) | she(1) | | |
| Gender Generation 3 | | | | | |
| baron(10) | lad(10) | uncle(10) | nephew(10) | lord(10) | aunt(9) |
| king(9) | knight(9) | protector(9) | belle(9) | lady(8) | bride(8) |
| matron(8) | gentleman(8) | bachelor(8) | godfather(8) | duchess(8) | princess(8) |
| chap(8) | youth(8) | queen(7) | hero(7) | groomsman(7) | matriarch(7) |
| empress(7) | hostess(7) | squire(7) | heroine(7) | mother(6) | sister(6) |
| buddy(6) | dame(6) | duke(6) | daughter-in-law(6) | countess(6) | prince(6) |
| boy(5) | brother(5) | madame(5) | niece(5) | maid(5) | groom(5) |
| motherhood(5) | elder(5) | master(5) | sister-in-law(5) | mother-in-law(5) | damsel(5) |
| vixen(5) | best man(5) | father(4) | daughter(4) | grandfather(4) | junior(4) |
| stag(4) | bachelorette(4) | bestie(4) | sir(4) | son(4) | boyfriend(4) |
| count(4) | heir(4) | host(4) | Pa(4) | gal(4) | mummy(4) |
| bridesmaid(4) | Miss(4) | maternal(4) | he(3) | mister(3) | girlfriend(3) |
| granddaughter(3) | brother-in-law(3) | sire(3) | goddess(3) | son-in-law(3) | patriarch(3) |
| lass(3) | mom(3) | mama(3) | grandmother(3) | baroness(3) | missus(3) |
| grandson(3) | girl(2) | female(2) | husband(2) | Papa(2) | wife(2) |
| maiden(2) | guy(2) | male(2) | man(1) | she(1) | Mrs.(1) |
| dad(1) | feminine(1) | woman(1) | emperor(1) | godmother(1) | gentlewoman(1) |
| Ma(1) | Mr.(1) | | | | |

## 22 Appendix - Subject words for General Prompting Data

Table 11: Subject Words for Racial Bias Data through Targeted Prompting

| | | Race Generation 1 | | | |
|---|---|---|---|---|---|
| arab(6) | melanesian(6) | ethiopian(6) | filipino(6) | malay(6) | basque(6) |
| icelander(6) | dutch(6) | serbian(6) | bengali(6) | scottish(5) | turkish(5) |
| japanese(5) | korean(5) | persian(5) | italian(5) | french(5) | native american(5) |
| maori(5) | ashkenazi(5) | slavic(5) | thai(5) | vietnamese(5) | kurd(5) |
| yoruba(5) | zulu(5) | hausa(5) | somali(5) | romani(5) | catalan(5) |
| greek(5) | norwegian(5) | finnish(5) | polish(5) | hungarian(5) | kosovar(5) |
| armenian(5) | uzbek(5) | kyrgyz(5) | tajik(5) | sinhalese(5) | khmer(5) |
| bantu(5) | guarani(5) | quechua(5) | aymara(5) | latino(4) | latina(4) |
| african(4) | european(4) | chinese(4) | indian(4) | russian(4) | german(4) |
| irish(4) | australian aboriginal(4) | polynesian(4) | jewish(4) | pacific islander(4) | berber(4) |
| pashtun(4) | igbo(4) | danish(4) | swiss(4) | portuguese(4) | bulgarian(4) |
| ukrainian(4) | belarusian(4) | croatian(4) | bosniak(4) | macedonian(4) | albanian(4) |
| georgian(4) | azerbaijani(4) | kazakh(4) | punjabi(4) | burmese(4) | javanese(4) |
| sundanese(4) | malagasy(4) | maltese(4) | sami(4) | inuit(4) | sherpa(4) |
| yazidi(4) | hispanic(3) | sephardi(3) | baltic(3) | xhosa(3) | swedish(3) |
| belgian(3) | romanian(3) | moldovan(3) | tamil(3) | lao(3) | creole(3) |
| tatar(3) | tibetan(3) | druze(3) | sunni(3) | ainu(3) | oromo(3) |
| bedouin(3) | samoan(3) | kikuyu(3) | white(2) | asian(2) | tuareg(2) |
| czech(2) | slovak(2) | montenegrin(2) | turkmen(2) | black(3) | aleut(2) |
| uighur(2) | maronite(2) | alawite(2) | maasai(2) | welsh(2) | chamorro(2) |
| mestizo(1) | bashkir(1) | nepali(1) | micronesian(1) | fijian(1) | tongan(1) |
| hawaiian(1) | latvian(1) | nenets(1) | mexican(1) | maldivian(1) | bosnian(1) |
| estonian(1) | | | | | |
| | | Race Generation 2 | | | |
| ashkenazi(6) | scottish(5) | turkish(5) | latino(5) | african(5) | european(5) |
| japanese(5) | korean(5) | arab(5) | persian(5) | italian(5) | french(5) |
| native american(5) | maori(5) | polynesian(5) | melanesian(5) | ethiopian(5) | slavic(5) |
| filipino(5) | thai(5) | vietnamese(5) | malay(5) | berber(5) | pashtun(5) |
| igbo(5) | yoruba(5) | zulu(5) | somali(5) | romani(5) | greek(5) |
| norwegian(5) | finnish(5) | dutch(5) | swiss(5) | portuguese(5) | bulgarian(5) |
| bosniak(5) | macedonian(5) | georgian(5) | kazakh(5) | punjabi(5) | malagasy(5) |
| bantu(5) | aymara(5) | yazidi(5) | hispanic(4) | chinese(4) | german(4) |
| irish(4) | sephardi(4) | pacific islander(4) | tuareg(4) | catalan(4) | danish(4) |
| icelander(4) | belgian(4) | slovak(4) | hungarian(4) | kosovar(4) | armenian(4) |
| azerbaijani(4) | uzbek(4) | kyrgyz(4) | tajik(4) | bengali(4) | sinhalese(4) |
| burmese(4) | khmer(4) | lao(4) | javanese(4) | sundanese(4) | maltese(4) |
| guarani(4) | quechua(4) | inuit(4) | bedouin(4) | chamorro(4) | ainu(4) |
| indian(3) | russian(3) | australian aboriginal(3) | jewish(3) | kurd(3) | hausa(3) |
| swedish(3) | czech(3) | ukrainian(3) | belarusian(3) | croatian(3) | serbian(3) |
| montenegrin(3) | albanian(3) | moldovan(3) | tamil(3) | sami(3) | hawaiian(3) |
| tongan(3) | druze(3) | sherpa(3) | mestizo(3) | chukchi(3) | micronesian(3) |
| bashkir(3) | khoisan(3) | fijian(3) | samoan(3) | black(2) | white(2) |
| asian(2) | latina(2) | baltic(2) | xhosa(2) | basque(2) | polish(2) |
| romanian(2) | turkmen(2) | maasai(2) | kikuyu(2) | oromo(2) | maronite(2) |
| kurdish(2) | creole(2) | tatar(2) | uighur(2) | tibetan(2) | nepali(2) |
| alawite(2) | tuvaluan(2) | welsh(1) | aleut(1) | mulatto(1) | chuvash(1) |
| shia(1) | sunni(1) | shona(1) | mandinka(1) | fulani(1) | nenets(1) |
| yakut(1) | icelandic(1) | mexican(1) | bosnian(1) | | |
| | | Race Generation 3 | | | |
| persian(7) | vietnamese(6) | armenian(6) | french(5) | japanese(5) | scottish(5) |
| african(5) | kurd(5) | italian(5) | bantu(5) | turkish(5) | ashkenazi(5) |
| hispanic(5) | yoruba(5) | korean(5) | arab(5) | quechua(5) | romani(5) |
| chinese(5) | indian(5) | kazakh(5) | macedonian(5) | bedouin(5) | azerbaijani(5) |
| ukrainian(5) | slavic(5) | german(5) | sherpa(5) | greek(5) | pashtun(5) |
| sephardi(5) | khmer(5) | swedish(5) | belarusian(5) | serbian(5) | javanese(5) |
| lao(5) | bosniak(5) | maltese(5) | kyrgyz(5) | latino(5) | ethiopian(5) |
| bengali(5) | thai(5) | georgian(5) | latina(5) | dutch(5) | finnish(5) |
| sinhalese(5) | maori(4) | native american(4) | inuit(4) | jewish(4) | polynesian(4) |
| icelander(4) | bulgarian(4) | somali(4) | european(4) | pacific islander(4) | basque(4) |
| norwegian(4) | zulu(4) | catalan(4) | tajik(4) | maasai(4) | hawaiian(4) |
| yazidi(4) | irish(4) | chamorro(4) | kikuyu(4) | samoan(4) | polish(4) |
| burmese(4) | igbo(4) | belgian(4) | kosovar(4) | portuguese(4) | moldovan(4) |
| guarani(4) | melanesian(4) | filipino(4) | russian(4) | albanian(4) | malagasy(4) |
| tongan(3) | aymara(3) | oromo(3) | tatar(3) | nenets(3) | croatian(3) |
| malay(3) | micronesian(3) | ainu(3) | punjabi(3) | sami(3) | hausa(3) |
| australian aboriginal(3) | danish(3) | czech(3) | khoisan(3) | uzbek(3) | sundanese(3) |
| druze(2) | fijian(2) | bashkir(2) | uighur(2) | tuvaluan(2) | baltic(2) |
| brazilian(2) | estonian(2) | creole(2) | swiss(2) | aleut(2) | montenegrin(2) |
| black(3) | slovak(2) | turkmen(2) | tamil(2) | mestizo(2) | fulani(1) |
| berber(1) | chukchi(1) | tibetan(1) | icelandic(1) | cuban(1) | maldivian(1) |
| palestinian(1) | mongolian(1) | tuareg(1) | bolivian(1) | kurdish(1) | slovakian(1) |
| bosnian(1) | xhosa(1) | hungarian(1) | romanian(1) | mulatto(1) | chuvash(1) |
| white(1) | asian(1) | welsh(1) | | | |

# 23 Appendix - Attribute words for Targeted Prompting Data

Table 12: Subject Words for Religious Bias Data through Targeted Prompting

| | | | | | |
|---|---|---|---|---|---|
| **Religion Generation 1** | | | | | |
| analects(9) | druidry(9) | voodoo(8) | torah(7) | guru granth sahib(7) | shamanism(7) |
| zen(7) | sufism(7) | gospel(7) | talmud(6) | taoism(6) | baha'i(6) |
| book of mormon(6) | rastafarianism(6) | wicca(6) | santeria(6) | mahayana(6) | kabbalah(6) |
| hasidism(6) | yazidism(6) | deism(6) | pantheism(6) | unitarianism(6) | mennonite(6) |
| mosque(6) | church(6) | tao te ching(6) | kitáb-i-aqdas(6) | alevism(6) | avesta(6) |
| shinto(6) | candomblé(6) | vajrayana(6) | druze(6) | quran(5) | buddhism(5) |
| christian(5) | jainism(5) | sikhism(5) | hadith(5) | catholic(5) | orthodox(5) |
| paganism(5) | native american church(5) | falun gong(5) | dianetics(5) | theravada(5) | coptic(5) |
| gnosticism(5) | monotheism(5) | presbyterianism(5) | amish(5) | jehovah's witnesses(5) | synagogue(5) |
| temple(5) | monastery(5) | ritual(5) | bektashi(5) | agnosticism(5) | atheism(5) |
| animism(5) | nichiren(5) | wahhabism(5) | ahmadiyya(5) | calvinism(5) | seventh-day adventist(5) |
| society of friends(5) | universalism(5) | dualism(5) | baptism(5) | hindu(5) | protestant(4) |
| zoroastrianism(4) | kojiki(4) | lutheran(4) | pilgrimage(4) | umbanda(4) | samaritanism(4) |
| polytheism(4) | manichaeism(4) | anglicanism(4) | church of satan(4) | tenrikyo(4) | bible(4) |
| mandaeanism(4) | islam(3) | shia(3) | quakerism(3) | scientology(3) | sunni(3) |
| mormonism(3) | confucianism(3) | upanishads(2) | lutheranism(1) | pagan(1) | centers(1) |
| puranas(1) | tantrism(1) | bhagavad gita(1) | hare krishna(1) | shaktism(1) | vaishnavism(1) |
| shaivism(1) | sankhya(1) | vedanta(1) | advaita(1) | rigveda(1) | samaveda(1) |
| atharvaveda(1) | brahmanas(1) | aranyakas(1) | | | |
| **Religion Generation 2** | | | | | |
| baha'i(9) | candomblé(9) | wicca(8) | sufism(8) | jainism(7) | talmud(7) |
| protestant(7) | zoroastrianism(7) | kojiki(7) | tao te ching(7) | analects(7) | kitáb-i-aqdas(7) |
| voodoo(7) | animism(7) | paganism(7) | druidry(7) | shamanism(7) | church of satan(7) |
| mosque(7) | bible(6) | hadith(6) | orthodox(6) | avesta(6) | taoism(6) |
| rastafarianism(6) | nichiren(6) | zen(6) | kabbalah(6) | gospel(6) | synagogue(6) |
| hindu(5) | quran(5) | buddhism(5) | torah(5) | christian(5) | sikhism(5) |
| guru granth sahib(5) | islam(5) | sunni(5) | shia(5) | catholic(5) | shinto(5) |
| confucianism(5) | mormonism(5) | book of mormon(5) | santeria(5) | umbanda(5) | native american church(5) |
| samaritanism(5) | tenrikyo(5) | theravada(5) | mahayana(5) | vajrayana(5) | wahhabism(5) |
| ahmadiyya(5) | coptic(5) | gnosticism(5) | druze(5) | alevism(5) | bektashi(5) |
| deism(5) | polytheism(5) | universalism(5) | quakerism(5) | calvinism(5) | mennonite(5) |
| seventh-day adventist(5) | jehovah's witnesses(5) | scientology(5) | temple(5) | church(5) | monastery(5) |
| pilgrimage(5) | ritual(5) | mandaeanism(4) | falun gong(4) | dianetics(4) | hasidism(4) |
| yazidism(4) | agnosticism(4) | atheism(4) | pantheism(4) | monotheism(4) | dualism(4) |
| manichaeism(4) | unitarianism(4) | society of friends(4) | lutheran(4) | anglicanism(4) | presbyterianism(4) |
| amish(4) | baptism(4) | atheist(1) | agnostic(1) | churches(1) | |
| **Religion Generation 3** | | | | | |
| wicca(10) | voodoo(8) | zen(8) | sufism(8) | jainism(7) | guru granth sahib(7) |
| kabbalah(7) | gospel(7) | church(7) | torah(6) | talmud(6) | hadith(6) |
| taoism(6) | tao te ching(6) | confucianism(6) | analects(6) | book of mormon(6) | rastafarianism(6) |
| animism(6) | paganism(6) | church of satan(6) | vajrayana(6) | hasidism(6) | coptic(6) |
| gnosticism(6) | druze(6) | yazidism(6) | alevism(6) | atheism(6) | deism(6) |
| pantheism(6) | manichaeism(6) | unitarianism(6) | universalism(6) | calvinism(6) | lutheran(6) |
| presbyterianism(6) | mennonite(6) | scientology(6) | temple(6) | pilgrimage(6) | ritual(6) |
| quran(5) | buddhism(5) | christian(5) | catholic(5) | orthodox(5) | avesta(5) |
| kojiki(5) | baha'i(5) | kitáb-i-aqdas(5) | druidry(5) | shamanism(5) | santeria(5) |
| candomblé(5) | umbanda(5) | mandaeanism(5) | falun gong(5) | dianetics(5) | tenrikyo(5) |
| nichiren(5) | theravada(5) | mahayana(5) | wahhabism(5) | ahmadiyya(5) | bektashi(5) |
| agnosticism(5) | polytheism(5) | monotheism(5) | dualism(5) | society of friends(5) | anglicanism(5) |
| amish(5) | baptism(5) | seventh-day adventist(5) | synagogue(5) | mosque(5) | monastery(5) |
| hindu(4) | sikhism(4) | sunni(4) | protestant(4) | shinto(4) | mormonism(4) |
| native american church(4) | samaritanism(4) | jehovah's witnesses(4) | bible(4) | islam(3) | shia(3) |
| zoroastrianism(3) | quakerism(3) | presbyterian(1) | | | |

# 24 Appendix - Attribute words for General Prompting Data

Table 13: Subject Words for General Prompting Data

| General Generation 1 | | | | | |
|---|---|---|---|---|---|
| librarian(5) | nun(4) | ceo(4) | rapper(4) | biker(4) | accountant(4) |
| lawyer(4) | bartender(4) | bodybuilder(4) | punk(4) | skateboarder(4) | desert(4) |
| boxer(4) | politician(4) | model(4) | tattoos(3) | football player(3) | africa(3) |
| hijab(3) | truck driver(3) | petite(3) | monk(3) | janitor(3) | soldier(3) |
| comedian(3) | mechanic(3) | butcher(3) | software engineer(3) | wrestler(3) | carpenter(3) |
| physicist(3) | mathematician(3) | men(3) | fisherman(3) | pilot(3) | farmer(3) |
| baker(3) | age(2) | teenager(2) | wealthy(2) | city(2) | scientist(2) |
| muscular(2) | gamer(2) | beauty queen(2) | construction worker(2) | rugby player(2) | actor(2) |
| surfer(2) | firefighter(2) | prison guard(2) | cowboy(2) | goth(2) | cab driver(2) |
| basketball player(2) | cheerleader(2) | slums(2) | banker(2) | athlete(2) | judge(2) |
| chef(2) | rocker(2) | insurance agent(2) | seamstress(2) | architect(2) | detective(2) |
| surgeon(2) | journalist(2) | teacher(2) | fishermen(2) | gamers(2) | asian(2) |
| australian(2) | arctic(2) | blonde(2) | fashionista(2) | dancer(2) | sailor(2) |
| astronaut(2) | tattoo artist(2) | flight attendant(2) | barista(2) | drummer(2) | cashier(2) |
| plumber(2) | wall street(2) | priest(2) | coal(2) | fireman(2) | male(1) |
| woman(1) | asians(1) | blind(1) | tech valley(1) | immigrant(1) | disability(1) |
| traditional(1) | overweight(1) | fashion model(1) | height(1) | housewife(1) | police officer(1) |
| astrophysicist(1) | millionaire(1) | sumo wrestler(1) | hip-hop artist(1) | saleswoman(1) | princess(1) |
| developer(1) | powerlifter(1) | motorcyclist(1) | metalworker(1) | security guard(1) | tattooed(1) |
| vet(1) | manager(1) | miner(1) | consultant(1) | podiatrist(1) | engineer(1) |
| radiologist(1) | bus driver(1) | painter(1) | receptionist(1) | anaesthesiologist(1) | engineers(1) |
| football team(1) | politicians(1) | dancers(1) | grandparents(1) | bodybuilders(1) | chefs(1) |
| writers(1) | farmers(1) | fashion models(1) | construction workers(1) | software developers(1) | musicians(1) |
| artists(1) | lawyers(1) | mathematicians(1) | firemen(1) | economists(1) | rugby players(1) |
| soldiers(2) | business executives(1) | doctors(1) | teenagers(2) | philosophers(1) | teachers(1) |
| truck drivers(1) | pilots(1) | nurses(1) | architects(1) | astronauts(1) | veterinarians(1) |
| bankers(1) | actors(1) | journalists(1) | children(2) | elderly women(1) | carpenters(1) |
| marathon runners(1) | boxers(1) | bakers(1) | plumbers(1) | electricians(1) | accountants(1) |
| dentists(1) | sailors(1) | florists(1) | mail carriers(1) | singers(1) | zoologists(1) |
| waiters(1) | skaters(1) | swimmers(1) | poets(1) | tax consultants(1) | ranchers(1) |
| gardeners(1) | hairdressers(1) | janitors(1) | painters(1) | mechanics(1) | taxi drivers(1) |
| gymnasts(1) | comedians(1) | surgeons(1) | cooks(1) | photographers(1) | real estate agents(1) |
| salespeople(1) | welders(1) | butchers(1) | basketball players(1) | barbers(1) | security guards(1) |
| theatre actors(1) | mixologists(1) | tailors(1) | optometrists(1) | veterans(1) | beekeepers(1) |
| shopkeepers(1) | metalworkers(1) | dog trainers(1) | housekeepers(1) | cyclists(1) | bricklayers(1) |
| rappers(1) | volleyball players(1) | podcasters(1) | cleaners(1) | farm workers(1) | tattoo artists(1) |
| cinematographers(1) | cosmetologists(1) | mountain climbers(1) | bartenders(1) | police officers(1) | 70(1) |
| middle east(1) | heavy build(1) | countryside(1) | conservative(1) | americans(1) | wheelchair(1) |
| urban(1) | hipster(1) | nerdy(1) | introvert(1) | gothic(1) | italians(1) |
| businessman(1) | glasses(1) | tropical island(1) | india(1) | plains(1) | young(1) |
| brazil(1) | dj(1) | actress(1) | snowy(1) | russian(1) | british(1) |
| metal artist(1) | germans(1) | policeman(1) | graffiti artist(1) | hairdresser(1) | spaniards(1) |
| iceland(1) | magician(1) | french(1) | dentist(1) | mexicans(1) | techie(1) |
| pageant queen(1) | scandinavians(1) | mma fighter(1) | kindergarten teacher(1) | appalachia(1) | football(1) |
| elderly(1) | construction(1) | vegan(1) | south america(1) | inner city(1) | software(1) |
| monastery(1) | visually impaired(1) | tribe(1) | ballerina(1) | homeless(1) | bronx(1) |
| metal(1) | tech(1) | luxury(1) | rural(1) | texas(1) | waitress(1) |
| island(1) | japan(1) | hollywood(1) | jazz(1) | weightlifter(1) | mountains(1) |
| sprinter(1) | corporate(1) | basketball(1) | beverly hills(1) | trucker(1) | midwest(1) |
| amazon(1) | sahara(1) | silicon valley(1) | sumo(1) | pro-gamer(1) | cop(1) |
| greenland(1) | opera(1) | tropical(1) | himalayas(1) | snowboarder(1) | lion(1) |
| alaska(1) | florist(1) | diver(1) | fashion(1) | tokyo(1) | salsa(1) |
| tattoo(1) | fighter(1) | rock star(1) | grandmother(1) | punk rocker(1) | principal(1) |
| nurse(1) | guitarist(1) | climber(1) | taxi driver(1) | chemist(1) | vlogger(1) |
| lifeguard(1) | hockey player(1) | hygienist(1) | conductor(1) | news anchor(1) | mailman(1) |
| veterinarian(1) | curator(1) | opera singer(1) | bouncer(1) | dietician(1) | radio jockey(1) |
| psychic(1) | historian(1) | real estate agent(1) | zookeeper(1) | sound engineer(1) | chiropractor(1) |
| flight instructor(1) | welder(1) | racecar driver(1) | hotel manager(1) | foreman(1) | marine biologist(1) |
| stuntman(1) | pianist(1) | video game developer(1) | electrician(1) | sheriff(1) | stockbroker(1) |
| photojournalist(1) | | | | | |

Table 14: Subject Words for General Prompting Data

| General Generation 2 | | | | | |
|---|---|---|---|---|---|
| construction worker(5) | security guard(5) | farmer(5) | janitor(5) | plumber(5) | cheerleader(4) |
| truck driver(4) | barista(4) | boxer(4) | librarian(4) | fisherman(4) | hairdresser(4) |
| accountant(4) | hip-hop artist(4) | dj(4) | bodybuilder(4) | waitress(4) | gamer(3) |
| mechanic(3) | rapper(3) | flight attendant(3) | metalhead(3) | florist(3) | cab driver(3) |
| principal(3) | firefighter(3) | punk rocker(3) | pastry chef(3) | banker(3) | fashion designer(3) |
| zookeeper(3) | cashier(3) | tattoo artist(3) | lifeguard(3) | butcher(3) | clown(3) |
| bartender(3) | rugby player(2) | athlete(2) | ballet dancer(2) | biker(2) | countryside(2) |
| kindergarten teacher(2) | teenager(2) | soldier(2) | businessman(2) | software engineer(2) | politician(2) |
| wrestler(2) | hipster(2) | goth(3) | chef(2) | beauty queen(2) | cop(2) |
| stay-at-home mom(2) | receptionist(2) | surgeon(2) | football player(2) | artist(2) | dentist(2) |
| housekeeper(2) | bus driver(2) | electrician(2) | car mechanic(2) | veterinarian(2) | model(2) |
| ceo(2) | bikers(2) | rappers(2) | farmers(3) | elderly(2) | ceos(2) |
| inner city(2) | nun(2) | bouncer(2) | actress(2) | fast-food worker(2) | maid(2) |
| firefighters(2) | sumo wrestler(2) | wheelchair(2) | surfer(2) | valet(2) | preschool teacher(2) |
| gardener(2) | window washer(2) | intern(2) | stuntman(2) | custodian(2) | tailor(2) |
| graffiti artist(2) | supermodel(2) | drummer(2) | hijab(2) | taxi driver(2) | mma fighter(2) |
| monk(2) | pop star(2) | fashionista(2) | nail technician(2) | bricklayer(2) | miner(2) |
| street vendor(2) | shepherd(2) | monks(2) | father(1) | older adult(1) | tattooed man(1) |
| female developer(1) | physique(1) | millennial(1) | fashion model(1) | young girl(1) | celebrity(1) |
| financial broker(1) | elderly woman(1) | rock musician(1) | female soccer player(1) | dropout(1) | gothic girl(1) |
| hollywood actor(1) | punk(1) | salesman(1) | introvert(1) | elderly gentleman(1) | real estate agent(1) |
| scientist(1) | young boy(1) | architect(1) | hedge fund manager(1) | lawyer(1) | office clerk(1) |
| math teacher(1) | corporate executive(1) | butler(1) | hair stylist(1) | pilot(1) | marine biologist(1) |
| neuroscientist(1) | beautician(1) | military general(1) | history professor(1) | tax consultant(1) | personal trainer(1) |
| data analyst(1) | grandma(1) | footballers(2) | blondes(1) | men(2) | women(2) |
| children(1) | teenagers(2) | tech geek(1) | athletes(1) | country singers(1) | cheerleaders(2) |
| goths(1) | homeless(1) | skaters(1) | rockstars(1) | soldiers(1) | gamers(2) |
| investment banker(1) | mime(1) | delivery guy(1) | astronaut(1) | flight instructor(1) | paparazzo(1) |
| retail worker(1) | gas station attendant(1) | car salesman(1) | dog walker(1) | telemarketer(1) | grocery store clerk(1) |
| carnival worker(1) | pool cleaner(1) | shoe shiner(1) | night watchman(1) | train conductor(1) | octogenarian(1) |
| stockbroker(1) | sari(1) | skateboarder(1) | cowboy(1) | hollywood(1) | gang member(1) |
| sorority(1) | lumberjack(1) | navy seal(1) | driver(1) | goalkeeper(1) | figure skater(1) |
| attorney(1) | officer(1) | milkman(1) | garbage collector(1) | postman(1) | gravedigger(1) |
| babysitter(1) | bellboy(1) | delivery man(1) | seamstress(1) | shop assistant(1) | baker(1) |
| shoemaker(1) | shoeshiner(1) | player(1) | winemaker(1) | boys(1) | older employees(1) |
| western tourists(1) | male harpists(1) | african(1) | introverts(1) | young children(1) | asian poets(1) |
| blind(1) | american tourists(1) | male authors(1) | deaf(1) | female engineers(1) | rural(1) |
| urban dwellers(1) | immigrants(1) | skateboarders(1) | muslim women(1) | older generation(1) | overweight(1) |
| dancer(1) | tattooed(2) | locals(1) | artists(1) | tribes(1) | tech enthusiasts(1) |
| bodybuilders(1) | latin american(1) | entrepreneurs(1) | librarians(1) | truck drivers(1) | people with disabilities(1) |
| vegetarians(1) | homeless man(1) | fashion designers(1) | priests(1) | refugees(1) | veterans(1) |
| metal musician(1) | lower economic backgrounds(1) | residents(1) | cat lovers(1) | dog enthusiasts(1) | models(1) |
| lawyers(1) | aristocrats(1) | computer programmers(1) | grandmothers(1) | golfers(1) | policemen(1) |
| bankers(1) | bakers(1) | heavy metal fans(1) | politicians(1) | mechanics(1) | construction workers(1) |
| waitresses(1) | wrestlers(1) | elders(1) | chefs(1) | accountants(1) | hairdressers(1) |
| janitors(1) | taxi drivers(1) | doorman(1) | clowns(1) | martial artists(1) | nurses(1) |
| pilots(1) | painters(1) | electricians(1) | fishermen(1) | rugby players(1) | djs(1) |
| opera singers(1) | jewelers(1) | ice cream vendor(1) | cinematographers(1) | senior(1) | jane(1) |
| abdullah(1) | young(1) | muscular(1) | biker gang(1) | jazz musician(1) | tribal(1) |
| punk rock singer(1) | sailor(1) | auto-rickshaw driver(1) | prima donna(1) | drag(1) | slums(1) |
| prison bars(1) | heavy metal guitarist(1) | data analysis(1) | army(1) | oil rig worker(1) | bedouin(1) |
| royal family(1) | coal mine(1) | rodeo cowboy(1) | village(1) | high heels(1) | tribal woman(1) |
| professional wrestler(1) | factory worker(1) | skateboarding(1) | snowboarder(1) | stiletto-clad(1) | circus acrobat(1) |
| rickshaw puller(1) | cosplayer(1) | nurse(1) | mail(1) | basketball player(1) | nightclub singer(1) |
| ballerina(1) | factory supervisor(1) | e-sports champion(1) | | | |

Table 15: Subject Words for General Prompting Data

| General Generation 3 | | | | | |
|---|---|---|---|---|---|
| bodybuilder(6) | janitor(5) | tattoo artist(5) | accountant(5) | mechanic(5) | surfer(5) |
| rapper(5) | boxer(5) | librarian(5) | butcher(5) | truck driver(4) | dancer(4) |
| city(4) | farmer(4) | biker(4) | taxi driver(4) | construction worker(4) | sailor(4) |
| skateboarder(4) | software developer(4) | banker(4) | carpenter(4) | bartender(4) | firefighter(4) |
| ceo(3) | fashion model(3) | basketball player(3) | gamer(3) | martial artist(3) | detective(3) |
| politician(3) | electrician(3) | teenager(3) | chef(3) | plumber(3) | flight attendant(3) |
| actor(3) | gardener(3) | cheerleader(3) | barista(3) | graffiti artist(3) | corporate(3) |
| fisherman(3) | nun(3) | male(2) | rural(2) | grandmother(2) | mathematician(2) |
| immigrants(2) | physicist(2) | linebacker(2) | opera singer(2) | comedian(2) | weightlifter(2) |
| pilot(2) | urban(2) | soldier(2) | motorcyclist(2) | scientist(2) | animator(2) |
| small town(2) | football player(2) | bikers(2) | cab driver(2) | mma fighter(2) | video gamer(2) |
| rock star(2) | monk(2) | punk rock(2) | beauty queen(2) | jazz musician(2) | pop singer(2) |
| hipster(2) | ghettos(2) | bellboy(2) | magician(2) | blonde(2) | hijab(2) |
| model(2) | wealthy(2) | housewife(2) | hairstylist(2) | miner(2) | postman(2) |
| baker(2) | receptionist(2) | lifeguard(2) | military(2) | coal miner(2) | desert(2) |
| kindergarten teacher(2) | mountain climber(2) | fashion(2) | sumo wrestler(2) | drummer(2) | fathers(1) |
| senior citizen(1) | men(1) | software engineer(1) | teenagers(1) | preschool teacher(1) | ceo's son(1) |
| introverts(1) | barber(1) | corporate lawyer(1) | boy(1) | mountains(1) | saleswoman(1) |
| millennial(1) | fireman(1) | biologist(1) | gymnast(1) | journalist(1) | dentist(1) |
| painter(1) | engineer(1) | soccer player(1) | editor(1) | neurosurgeon(1) | architect(1) |
| it specialist(1) | teacher(1) | nurse(1) | fitness instructor(1) | musician(1) | lawyer(1) |
| movie director(1) | programmer(1) | designer(1) | pharmacist(1) | office clerk(1) | veterinarian(1) |
| economist(1) | factory worker(1) | coach(1) | psychologist(1) | flight engineer(1) | podiatrist(1) |
| engineers(1) | projects(1) | managerial roles(1) | muscular(1) | frail(1) | fishermen(1) |
| lumberjack(1) | tech geek(1) | wrestler(1) | soldiers(1) | attorney(1) | miners(1) |
| wall street(1) | quarterback(1) | farm boy(1) | political leader(1) | heavyweight champion(1) | school teacher(1) |
| princess(1) | slums(1) | hip hop artist(1) | dj(1) | bouncer(1) | businessman(1) |
| actress(1) | hunters(1) | ballerina(1) | motorbike racer(1) | lawyers(1) | stock trader(1) |
| police officer(1) | comic book artist(1) | marine(1) | cabaret dancer(1) | hacker(1) | stuntman(1) |
| pop star(1) | nightlife(1) | circus performer(1) | rodeo(1) | ice hockey player(1) | adult films(1) |
| death metal singer(1) | dropout(1) | gangster(1) | paparazzo(1) | nomad(1) | televangelist(1) |
| stuntwoman(1) | pirates(1) | supermodel(1) | cage fighter(1) | race car drivers(1) | athletes(1) |
| elderly(1) | tattoos(1) | homeless(1) | introvert(1) | glasses(1) | footballer(1) |
| wheelchair(1) | rockstar(1) | goth(1) | skater(1) | tall(1) | mma(1) |
| slum(1) | fashionista(1) | truck(1) | punk(1) | nerd(1) | socialite(1) |
| grunge(1) | baseball(1) | policeman(1) | bus(1) | construction(1) | maid(1) |
| waitress(1) | cashier(1) | garbage(1) | florist(1) | security(1) | taxi(1) |
| pastry(1) | stewardess(1) | telemarketer(1) | custodian(1) | seamstress(1) | vet(1) |
| coal(1) | kindergarten(1) | factory(1) | milkman(1) | delivery(1) | mason(1) |
| store(1) | makeup(1) | street(1) | tailor(1) | masseuse(1) | fast-food(1) |
| gym(1) | nail(1) | cobbler(1) | groomer(1) | window(1) | attendant(1) |
| hygienist(1) | guard(1) | youth(1) | she(1) | elder(1) | he(1) |
| ireland(1) | mothers(1) | vegans(1) | christian(1) | visually impaired(1) | indigenous(1) |
| middle east(1) | monks(1) | tropics(1) | fishing(1) | landlocked(1) | amish(1) |
| ballet dancer(1) | football team(1) | ceo's(1) | tech(1) | heavy metal band(1) | rugby players(1) |
| conservative(1) | basketball(1) | farming(1) | inmates(1) | poverty(1) | cosmetics(1) |
| gang(1) | skyscrapers(1) | coal mines(1) | inner city(1) | plains(1) | physics teacher(1) |
| young girl(1) | traditional(1) | stockbroker(1) | reality tv star(1) | manicurist(1) | oil rig worker(1) |
| nail technician(1) | horse jockey(1) | candy store(1) | rock climber(1) | man(1) | aristocratic(1) |
| child(1) | wall street executive(1) | hip-hop artist(1) | war-torn region(1) | tattooed(1) | professional wrestler(1) |
| rugby player(1) | punk rocker(1) | ghetto(1) | heavy metal drummer(1) | tribal leader(1) | hip-hop dancer(1) |
| bus driver(1) | cowboy(1) | security guard(1) | astronaut(1) | horror writer(1) | judge(1) |
| metal worker(1) | race car driver(1) | prima donna(1) | street performer(1) | singer(1) | exterminator(1) |
| snowboarder(1) | mascot(1) | zookeeper(1) | bricklayer(1) | pastry chef(1) | swimmer(1) |
| stand-up comedian(1) | shopkeeper(1) | | | | |

Table 16: Attribute words for Gender Bias Data Through Targeted Prompting

| Gender Generation 1 | | | | | |
|---|---|---|---|---|---|
| delicate(7) | meticulous(7) | nurturing(6) | analytical(6) | tech-savvy(6) | gentle(5) |
| compassionate(5) | tenacious(5) | agile(5) | strategic(4) | innovative(4) | humble(4) |
| adventurous(4) | empathetic(4) | profound(4) | culinary(4) | prodigy(4) | fashion(3) |
| martial(3) | poetic(3) | leadership(3) | romantic(3) | driver(3) | robotics(3) |
| wise(3) | logical(3) | graceful(3) | audacious(3) | physicist(3) | empathy(2) |
| baking(2) | dance(2) | grounded(2) | physics(2) | ballet(2) | climbing(2) |
| weightlifting(2) | yoga(2) | action(2) | gourmet(2) | boxing(2) | video(2) |
| eloquent(2) | salsa(2) | pastry(2) | skincare(2) | virtuoso(2) | environmental(2) |
| emotional(2) | resourceful(2) | courageous(2) | protective(2) | shrewd(2) | calm(2) |
| patient(2) | cheerful(2) | mature(2) | imaginative(2) | attentive(2) | creative(2) |
| insightful(2) | skillful(2) | resilient(2) | humorous(2) | lively(2) | articulate(2) |
| candid(2) | jovial(2) | boisterous(2) | tactical(2) | intuitive(2) | whimsical(2) |
| flair(2) | sagacious(2) | voracious(2) | adept(2) | proficient(2) | astute(2) |
| erudite(2) | dexterous(2) | formidable(2) | brilliant(2) | artist(2) | entrepreneur(2) |
| mountaineer(2) | gardening(2) | dancer(2) | coder(2) | poet(2) | champion(2) |
| master(2) | warrior(2) | opera(2) | astrophysicist(2) | engineer(2) | astronomer(2) |
| architect(2) | marine(2) | athlete(2) | pilot(2) | biologist(2) | florist(2) |
| mechanic(2) | engineering(2) | stoic(1) | mechanical(1) | computer(1) | engine(1) |
| intuition(1) | commanding(1) | sew(1) | meditate(1) | historical(1) | music(1) |
| calligraphy(1) | astrophysics(1) | electronic(1) | aesthetic(1) | chess(1) | animation(1) |
| woodworking(1) | ornate(1) | sports(1) | pottery(1) | electric(1) | operatic(1) |
| basketball(1) | virtual(1) | graffiti(1) | code(1) | diving(1) | business(1) |
| violin(1) | detective(1) | ethereal(1) | punk(1) | architectural(1) | tech(1) |
| languages(1) | painting(1) | DJs(1) | mathematical(1) | bioengineering(1) | exploration(1) |
| flamenco(1) | blues(1) | skateboarder(1) | surreal(1) | AI(1) | sculpting(1) |
| artisanal(1) | finance(1) | conservation(1) | MMA(1) | laser(1) | sci-fi(1) |
| psychology(1) | lace(1) | compositions(1) | avant-garde(1) | encyclopedic(1) | mountaineering(1) |
| drummer(1) | floral(1) | textile(1) | acrobatics(1) | quantum(1) | theater(1) |
| barista(1) | archery(1) | soft-hearted(1) | determined(1) | cool-headed(1) | understanding(1) |
| laid-back(1) | fit(1) | powerful(1) | pragmatic(1) | fashionable(1) | open-minded(1) |
| thoughtful(1) | impeccable(1) | confident(1) | precise(1) | multitask(1) | energetic(1) |
| authoritative(1) | perceptive(1) | kind-hearted(1) | curious(1) | well-informed(1) | enthusiastic(1) |
| visionary(1) | level-headed(1) | expertise(1) | down-to-earth(1) | artistic(1) | muscular(1) |
| assertive(1) | comedic(1) | deep(1) | stern(1) | wiry(1) | detached(1) |
| brusque(1) | nonchalant(1) | sardonic(1) | flexibility(1) | trendy(1) | serene(1) |
| contemplative(1) | soft-spoken(1) | amiable(1) | frugal(1) | spontaneous(1) | infectious(1) |
| grace(1) | nimble(1) | phenomenal(1) | rambunctious(1) | adroit(1) | exquisite(1) |
| intrepid(1) | poignant(1) | discerning(1) | masterful(1) | deft(1) | robust(1) |
| prodigious(1) | nuanced(1) | resolute(1) | mellifluous(1) | vigorous(1) | lyrical(1) |
| fervent(1) | ebullient(1) | mesmerizing(1) | vivacious(1) | rugged(1) | strong(1) |
| ferocious(1) | groundbreaking(1) | athletic(1) | innovator(1) | tender-hearted(1) | genius(1) |
| environmentalist(1) | disciplined(1) | fiery(1) | philosophical(1) | simple(1) | eclectic(1) |
| tech-oriented(1) | progressive(1) | scientist(1) | quirky(1) | trailblazing(1) | musician(1) |
| botanist(1) | fierce(1) | comedian(1) | acumen(1) | photographer(1) | advocate(1) |
| humanitarian(1) | mathematician(1) | enthusiast(1) | geek(1) | philanthropist(1) | linguistics(1) |
| playwright(1) | climber(1) | historian(1) | painter(1) | neuroscience(1) | ecologist(1) |
| biomechanics(1) | sculptor(1) | pianist(1) | cryptography(1) | ceramist(1) | ornithologist(1) |
| economist(1) | geologist(1) | contemporary(1) | caregiver(1) | gentleness(1) | multitasking(1) |
| introspective(1) | cook(1) | support(1) | listener(1) | embroidery(1) | caring(1) |
| poetry(1) | tears(1) | resilience(1) | crafting(1) | classical(1) | arts(1) |
| rescue(1) | vulnerability(1) | style(1) | wisdom(1) | advocacy(1) | relate(1) |
| botany(1) | cars(1) | courage(1) | sword(1) | woodwork(1) | strength(1) |
| sharpshooter(1) | reptiles(1) | rugby(1) | breadwinner(1) | digital(1) | programming(1) |
| handyman(1) | electrical(1) | garden(1) | developers(1) | rocket(1) | blacksmith(1) |
| cyber(1) | rearing(1) | firefighter(1) | makeup(1) | cooking(1) | paintings(1) |
| Taekwondo(1) | pediatric(1) | race(1) | feminist(1) | | |

Table 17: Attribute words for Gender Bias Data Through Targeted Prompting

| Gender Generation 2 | | | | | |
|---|---|---|---|---|---|
| nurturing(8) | wisdom(8) | empathetic(7) | humble(8) | compassionate(6) | caring(7) |
| innovative(6) | ambitious(6) | resilient(6) | adventurous(6) | analytical(5) | down-to-earth(7) |
| wise(5) | independent(5) | tech-savvy(6) | strategic(5) | playful(5) | assertive(4) |
| introspective(5) | leadership(4) | sensitivity(4) | knowledgeable(3) | passionate(3) | sensitive(5) |
| audacious(4) | intuitive(4) | competitive(3) | understanding(3) | thinker(3) | bold(3) |
| protective(3) | vulnerability(3) | outspoken(4) | thoughtful(2) | kind(3) | articulate(2) |
| resourceful(2) | powerhouse(2) | sociable(2) | open-minded(2) | approachable(3) | brilliant(2) |
| protector(2) | leader(2) | advocate(2) | considerate(3) | genius(2) | grounded(2) |
| lover(2) | gamer(3) | athlete(3) | researcher(2) | entrepreneur(2) | logical(2) |
| expressive(2) | soft-spoken(2) | entrepreneurial(2) | affectionate(2) | pragmatic(3) | poetic(3) |
| intelligence(2) | gentle(4) | mature(3) | generous(2) | relatable(2) | attentive(2) |
| humorous(3) | committed(2) | insightful(2) | fun-loving(2) | intellectual(2) | witty(3) |
| audacity(2) | conservative(2) | wit(2) | stern(2) | empathy(2) | astute(3) |
| rugged(2) | boisterous(3) | lively(2) | goofy(3) | fashionable(3) | candid(2) |
| dancer(3) | humility(2) | helpful(1) | intelligent(2) | jovial(2) | talented(1) |
| diligent(1) | sharp(1) | curious(1) | friendly(1) | advisory(1) | loyal(1) |
| patient(2) | positive(1) | graceful(1) | listening(1) | risk-taker(1) | adaptable(1) |
| philanthropist(1) | comedian(1) | engineer(1) | champion(1) | trendsetter(1) | storyteller(1) |
| mingling(1) | economist(1) | chef(1) | scientist(1) | singer(2) | architect(1) |
| prodigy(1) | baker(1) | activist(1) | enthusiast(1) | connoisseur(1) | developer(1) |
| environmentalist(1) | educator(1) | karate(1) | novelist(1) | simple(1) | filmmaker(1) |
| well-read(1) | conservationist(1) | innovator(1) | historian(1) | poet(1) | climbing(1) |
| determined(1) | light-hearted(1) | eloquent(1) | hilarious(1) | worldly(1) | rational(1) |
| sentimental(2) | modest(2) | domestic(1) | authoritative(1) | feeling(1) | compassion(1) |
| tenacious(1) | stylish(1) | commanding(1) | strong(2) | listener(1) | fierce(2) |
| kind-hearted(2) | problem-solving(1) | joyful(1) | arrogant(2) | careless(1) | vulnerable(1) |
| shy(1) | introverted(2) | exceptional(1) | technological(1) | calm(2) | emotion(1) |
| submissive(1) | strategist(1) | inexperienced(1) | insecure(1) | anxious(1) | creative(1) |
| maternal(1) | whimsical(2) | flaws(1) | confident(1) | aloof(1) | tender(1) |
| non-serious(1) | selfless(1) | champions(1) | determination(1) | caregiving(1) | fashion(1) |
| adventure(1) | self-doubt(1) | stoic(1) | paternal(1) | sporty(1) | geek(1) |
| brains(1) | trendy(1) | modesty(1) | proactive(1) | domineering(1) | demeanor(1) |
| angry(1) | thin(1) | serious(1) | meek(1) | unassuming(1) | courageous(1) |
| rowdy(1) | silly(1) | frugal(1) | chatty(1) | bashful(1) | unpretentious(1) |
| giddy(1) | spunky(1) | informal(1) | delicate(1) | naive(1) | enthusiastic(1) |
| extroverted(1) | timid(1) | reflective(1) | cheeky(1) | tender-hearted(1) | laid-back(1) |
| old-soul(1) | expert(1) | nerdy(1) | cook(1) | sprightly(1) | zesty(1) |
| athletic(1) | voracious(1) | optimistic(1) | well-spoken(1) | sunny(1) | mechanical(1) |
| gardener(1) | mathematician(1) | painter(1) | patience(1) | brave(1) | lighthearted(1) |
| sharp-minded(1) | humor(1) | cries(1) | fiery(1) | diplomacy(1) | fighting(1) |
| laugh(1) | rebellious(1) | follow(1) | candidness(1) | tears(1) | values(1) |
| emotions(1) | daring(1) | peaceful(1) | transparent(1) | acknowledges(1) | quirkiness(1) |
| jokes(1) | arts(1) | party(1) | depth(1) | loyalty(1) | resilience(1) |
| romantic(1) | confrontations(1) | thinking(1) | vivacious(1) | mischievous(1) | competitor(1) |
| warrior(1) | supporting(1) | sharp-witted(1) | independence(1) | adventures(1) | distress(1) |
| generosity(1) | ground(1) | equality(1) | kindness(1) | strength(1) | guiding(1) |
| charm(1) | graciousness(1) | confidence(1) | caretaker(1) | mentor(1) | pleasures(1) |
| commitment(1) | approachability(1) | receptive(1) | tenacity(1) | | |

Table 18: Attribute words for Gender Bias Data Through Targeted Prompting

| Gender Generation 3 | | | | | |
|---|---|---|---|---|---|
| wise(23) | arrogant(22) | uncaring(22) | thin(21) | angry(19) | nurturing(5) |
| tech-savvy(5) | fashion(5) | fierce(4) | mechanic(4) | ballet(4) | playful(3) |
| naive(3) | wisdom(3) | modern(3) | humble(3) | compassionate(3) | humor(3) |
| tech(3) | prodigy(3) | physicist(3) | caring(2) | stern(2) | analytical(2) |
| dominant(2) | cook(2) | protector(2) | empathetic(2) | thoughtful(2) | thinker(2) |
| grace(2) | sensitive(2) | aloof(2) | life(2) | vulnerabilities(2) | rock(2) |
| supporter(2) | wild(2) | reader(2) | philosophical(2) | adventurer(2) | engineer(2) |
| dancer(2) | hero(2) | culinary(2) | resilient(2) | botanist(2) | mountaineer(2) |
| mathematics(2) | vegan(2) | climber(2) | driver(2) | robotics(2) | yoga(2) |
| biologist(2) | pastry(2) | advocate(2) | musician(2) | opera(2) | mogul(2) |
| novelist(2) | activist(2) | languages(2) | delicate(2) | jovial(2) | insightful(2) |
| poet(2) | wit(2) | gardener(2) | caregiver(2) | chess(2) | coding(2) |
| fat(2) | assertive(1) | logical(1) | discreet(1) | domesticated(1) | outspoken(1) |
| kind-hearted(1) | strategic(1) | cunning(1) | stoic(1) | mature(1) | committed(1) |
| fearless(1) | emotions(1) | rough(1) | collaborative(1) | resilience(1) | ruthless(1) |
| warriors(1) | frivolous(1) | serious(1) | jokester(1) | emotional(1) | peacemaker(1) |
| careless(1) | involved(1) | poets(1) | approachable(1) | deliberate(1) | responsible(1) |
| seeks(1) | admits(1) | extroverted(1) | listener(1) | meticulous(1) | open(1) |
| submissive(1) | scientist(1) | businesswoman(1) | breadwinner(1) | business(1) | politics(1) |
| competitive(1) | decisive(1) | gritty(1) | simplicity(1) | jester(1) | muscular(1) |
| baker(1) | knit(1) | coder(1) | poetic(1) | outpace(1) | repair(1) |
| astronomy(1) | soothing(1) | boxing(1) | artist(1) | gardening(1) | lawyer(1) |
| physics(1) | skateboarding(1) | potter(1) | astrophysicist(1) | zoologist(1) | calligraphy(1) |
| computer(1) | connoisseur(1) | neuroscientist(1) | writer(1) | grandmaster(1) | swimmer(1) |
| cellist(1) | cryptography(1) | comedy(1) | ornithology(1) | pilot(1) | fighter(1) |
| geneticist(1) | mentors(1) | saxophonist(1) | volcanologist(1) | sharpshooter(1) | linguistic(1) |
| developer(1) | architectural(1) | taekwondo(1) | skydiver(1) | ceramics(1) | photographer(1) |
| mathematician(1) | gourmet(1) | archeologist(1) | virtuoso(1) | biochemist(1) | astronaut(1) |
| skateboarder(1) | forensic(1) | perfumery(1) | artificial intelligence(1) | acrobatic(1) | archaeologist(1) |
| programming(1) | pianist(1) | neuroscience(1) | farming(1) | researcher(1) | patient(1) |
| lonely(1) | down-to-earth(1) | cold(1) | noble(1) | slender(1) | introspective(1) |
| gentle(1) | vulnerable(1) | timid(1) | kind(1) | determination(1) | vivacious(1) |
| generous(1) | fiery(1) | humility(1) | judgmental(1) | youthful(1) | adventurous(1) |
| reason(1) | grounded(1) | grateful(1) | elegance(1) | shine(1) | intellectual(1) |
| style(1) | intuitive(1) | artistic(1) | unapproachable(1) | corporate(1) | warmth(1) |
| connected(1) | confidante(1) | scholar(1) | substance(1) | ambition(1) | strategist(1) |
| genius(1) | mix(1) | archer(1) | confidence(1) | trend(1) | racer(1) |
| insights(1) | karate(1) | open-minded(1) | master(1) | rock-climbing(1) | boisterous(1) |
| self-sufficient(1) | storyteller(1) | maturity(1) | painting(1) | guitarist(1) | academic(1) |
| empathy(1) | minimalist(1) | expert(1) | renowned(1) | kindness(1) | cheerful(1) |
| engineering(1) | rescue(1) | environmentalist(1) | seasoned(1) | black belt(1) | comforting(1) |
| entrepreneur(1) | charity(1) | frugality(1) | brilliant(1) | championed(1) | singing(1) |
| charge(1) | dedication(1) | startup(1) | chef(1) | calmest(1) | eloquent(1) |
| botany(1) | architect(1) | compassion(1) | financial(1) | invention(1) | doctorate(1) |
| gentlest(1) | astrophysics(1) | authored(1) | rock climbing(1) | polymath(1) | teaches(1) |
| violinist(1) | comedian(1) | ace(1) | dance(1) | scuba diving(1) | watercolor(1) |
| florist(1) | wrestling(1) | marathon(1) | romance(1) | software(1) | ballroom(1) |
| martial arts(1) | comic(1) | story-telling(1) | woodwork(1) | bakes(1) | dj(1) |
| beekeeping(1) | weightlifting(1) | knitting(1) | gamer(1) | skydiving(1) | braids(1) |
| therapeutic(1) | gentleness(1) | pediatric(1) | rugby(1) | art(1) | makeup(1) |
| pottery(1) | carpentry(1) | adventure(1) | author(1) | salsa(1) | |

Table 19: Attribute words for Racial Bias Data Through Targeted Prompting

| | | **Race Generation 1** | | | |
|---|---|---|---|---|---|
| innovative(10) | groundbreaking(7) | spiritual(6) | profound(6) | harmonious(6) | musicians(6) |
| vibrant(5) | sustainable(5) | intricate(5) | educators(5) | precision(5) | introspective(4) |
| delightful(4) | enchanting(4) | unparalleled(4) | poets(4) | dancers(4) | environmentalists(4) |
| historians(4) | creativity(4) | resilience(4) | resilient(3) | soulful(3) | enlightening(3) |
| timeless(3) | pioneering(3) | meticulous(3) | lyrical(3) | artists(3) | filmmakers(3) |
| writers(3) | astronomers(3) | conservationists(3) | activists(3) | storytellers(3) | resourceful(3) |
| introspection(3) | craftsmanship(3) | respect(3) | unity(3) | wisdom(3) | poetic(3) |
| adaptability(3) | bravery(3) | progressive(3) | artistic(2) | visionary(2) | exceptional(2) |
| monumental(2) | holistic(2) | relentless(2) | mesmerizing(2) | transformative(2) | compassionate(2) |
| captivating(2) | adept(2) | ingenious(2) | flair(2) | vivid(2) | unique(2) |
| championing(2) | evocative(2) | entrepreneurs(2) | engineers(2) | architects(2) | playwrights(2) |
| farmers(2) | painters(2) | linguists(2) | biologists(2) | trailblazing(2) | dynamic(2) |
| discipline(2) | elegance(2) | strength(2) | harmony(2) | inclusivity(2) | valor(2) |
| innovations(2) | depth(2) | perseverance(2) | tranquility(2) | detailing(2) | courage(2) |
| essence(2) | warmth(2) | insightful(2) | vibrancy(2) | merge(2) | connection(2) |
| expanded(2) | revolutionary(2) | heartbeat(2) | philosophical(2) | adventurous(2) | tenacious(2) |
| literary(2) | rhythmic(2) | world-class(2) | astute(2) | contributed(2) | pushing(2) |
| adaptive(2) | indefatigable(2) | mesmerizes(2) | innovation(1) | integrated(1) | precise(1) |
| graceful(1) | pivotal(1) | passionate(1) | health-conscious(1) | committed(1) | heartwarming(1) |
| respectful(1) | unrivaled(1) | mysterious(1) | tireless(1) | seamless(1) | invaluable(1) |
| honorable(1) | raw(1) | courageous(1) | altruistic(1) | transcendent(1) | crucial(1) |
| connected(1) | determined(1) | fervent(1) | unquenchable(1) | steadfast(1) | embracing(1) |
| fresh(1) | unifying(1) | cutting-edge(1) | inspiring(1) | nuanced(1) | elegant(1) |
| energized(1) | resonant(1) | diverse(1) | unmatched(1) | welcoming(1) | dazzling(1) |
| reverent(1) | mindful(1) | awe-inspiring(1) | mythical(1) | stellar(1) | balanced(1) |
| knowledgeable(1) | innovators(1) | enriching(1) | imaginative(1) | leaders(1) | scholars(1) |
| designers(1) | chefs(1) | navigators(1) | philosophers(1) | researchers(1) | folklorists(1) |
| novelists(1) | ceramists(1) | sculptors(1) | ecologists(1) | journalists(1) | mathematicians(1) |
| technologists(1) | planners(1) | geologists(1) | chocolatiers(1) | watchmakers(1) | horticulturists(1) |
| photographers(1) | artisans(1) | scientists(1) | winemakers(1) | singers(1) | archaeologists(1) |
| crafters(1) | mountaineers(1) | puppeteers(1) | weavers(1) | herbalists(1) | herders(1) |
| shamans(1) | compassion(1) | self-awareness(1) | richness(1) | reliability(1) | wit(1) |
| eclectic(1) | solidarity(1) | joy(1) | ingenuity(1) | emotive(1) | exploration(1) |
| foresight(1) | endurance(1) | eloquent(1) | illumination(1) | brilliance(1) | festive(1) |
| critical-thinking(1) | wonder(1) | simplicity(1) | togetherness(1) | expertise(1) | trailblazers(1) |
| expressions(1) | imagination(1) | dedication(1) | serenity(1) | fellowship(1) | mosaic(1) |
| faith(1) | enthusiasm(1) | ties(1) | heritage(1) | humility(1) | balance(1) |
| melodic(1) | exchange(1) | understanding(1) | community(1) | fusion(1) | exhilarating(1) |
| honor(1) | symbolic(1) | detailed(1) | mindfulness(1) | devotion(1) | preservation(1) |
| tolerance(1) | revolutionized(1) | authenticity(1) | grace(1) | insights(1) | commitment(1) |
| exuberant(1) | enduring(1) | ecological(1) | passion(1) | valiant(1) | heartfelt(1) |
| boundless(1) | aesthetics(1) | genius(1) | soul-stirring(1) | mastery(1) | emotion(1) |
| hope(1) | bonds(1) | finesse(1) | oceanic(1) | delectable(1) | rhythm(1) |
| cosmic(1) | serene(1) | diversity(1) | admiration(1) | determination(1) | penned(1) |
| joyful(1) | perfection(1) | styles(1) | colors(1) | awe(1) | pulse(1) |
| texture(1) | hospitality(1) | shaped(1) | realm(1) | exuberance(1) | realms(1) |
| resonance(1) | landscapes(1) | arctic(1) | tranquil(1) | heart(1) | mystic(1) |
| delights(1) | mirror(1) | shine(1) | cosmos(1) | epitomize(1) | dazzle(1) |
| versatility(1) | astuteness(1) | linguistic(1) | intellectual(1) | resourcefulness(1) | pioneers(1) |
| analytical(1) | trustworthy(1) | entrepreneurial(1) | rich(1) | reflective(1) | legendary(1) |
| trendsetting(1) | finest(1) | architectural(1) | versatile(1) | indomitable(1) | enriched(1) |
| influential(1) | | | | | |

Table 20: Attribute words for Racial Bias Data Through Targeted Prompting

| Race Generation 2 | | | | | |
|---|---|---|---|---|---|
| artistic(29) | cultural(26) | historical(19) | diverse(17) | intellectual(10) | scientific(10) |
| sustainable(8) | vibrant(6) | culinary(6) | pioneering(5) | intricate(5) | innovative(5) |
| incorporate(5) | instrumental(4) | harmonious(4) | soulful(3) | profound(3) | renewable(3) |
| revolutionary(3) | meticulous(3) | evocative(3) | vivacious(3) | precision(3) | unity(3) |
| excellent(3) | wisdom(3) | resilience(3) | introspection(3) | mesmerizing(2) | influential(2) |
| spiritual(2) | passionate(2) | contemporary(2) | championing(2) | holistic(2) | global(2) |
| contributed(2) | inspiration(2) | exploring(2) | gourmet(2) | draw(2) | inspired(2) |
| blend(2) | highlight(2) | fusion(2) | contributions(2) | groundbreaking(2) | resilient(2) |
| hospitable(2) | ingenious(2) | rooted(2) | enduring(2) | delightful(2) | universal(2) |
| poignant(2) | authentic(2) | acumen(2) | wise(2) | prowess(2) | cutting-edge(2) |
| reverence(2) | confluence(2) | tapestry(2) | literary(2) | navigational(2) | poetic(2) |
| modern(2) | ethical(2) | elegance(2) | avant-garde(2) | adaptability(2) | imaginative(2) |
| expertise(2) | forward-thinking(2) | creativity(2) | inventive(2) | dedication(2) | compassionate(1) |
| breaking(1) | renowned(1) | disciplined(1) | organic(1) | reimagining(1) | conservationist(1) |
| trendsetting(1) | admired(1) | utilize(1) | wisdom-filled(1) | magical(1) | appreciative(1) |
| blending(1) | inspire(1) | diving(1) | legendary(1) | experiment(1) | documented(1) |
| fantasy(1) | minimalistic(1) | recognized(1) | eclectic(1) | study(1) | mesmerized(1) |
| showcase(1) | connection(1) | merging(1) | fuse(1) | incorporated(1) | aesthetics(1) |
| muse(1) | liking(1) | resonance(1) | introduced(1) | penchant(1) | energy(1) |
| admiration(1) | preserve(1) | merge(1) | international(1) | masterpieces(1) | championed(1) |
| enthralling(1) | masterfully(1) | bring(1) | studied(1) | echo(1) | collaborate(1) |
| revolutionizing(1) | seamlessly(1) | crafting(1) | insightful(1) | creative(1) | accurate(1) |
| advanced(1) | eco-friendly(1) | original(1) | masterful(1) | integral(1) | judicious(1) |
| protective(1) | graceful(1) | tenacious(1) | enchanting(1) | stirring(1) | ethereal(1) |
| adapted(1) | lasting(1) | fearless(1) | dexterous(1) | forefront(1) | potent(1) |
| empowered(1) | cohesive(1) | mystical(1) | brilliant(1) | transcendent(1) | trailblazing(1) |
| sagacious(1) | serene(1) | relentless(1) | impeccable(1) | unified(1) | fervent(1) |
| marvelous(1) | sacred(1) | leading-edge(1) | dedicated(1) | skillful(1) | redefining(1) |
| niche(1) | mosaic(1) | unbroken(1) | helm(1) | knack(1) | zenith(1) |
| repository(1) | pushing(1) | finesse(1) | visionaries(1) | hauntingly(1) | delectable(1) |
| extraordinary(1) | resonate(1) | sanctity(1) | eloquent(1) | resonant(1) | balance(1) |
| inclusivity(1) | accomplished(1) | achievements(1) | significant(1) | engineering(1) | culturally(1) |
| academic(1) | compassion(1) | humanitarian(1) | philosophical(1) | inspiring(1) | nobility(1) |
| heartfelt(1) | conservation(1) | empathy(1) | solidarity(1) | reconciliation(1) | complexity(1) |
| philanthropic(1) | interconnectedness(1) | mysteries(1) | transformative(1) | heritage(1) | contemplative(1) |
| community(1) | justice(1) | joy(1) | timeless(1) | romance(1) | grace(1) |
| wildlife(1) | illuminating(1) | restore(1) | exquisite(1) | dialogue(1) | perspectives(1) |
| spotlight(1) | sanctuary(1) | lyrical(1) | mesmerize(1) | foundation(1) | advocating(1) |
| unique(1) | progressive(1) | joyful(1) | scholarly(1) | empathetic(1) | romanticism(1) |
| eloquence(1) | daring(1) | astuteness(1) | harmony(1) | industrious(1) | keen(1) |
| research(1) | intellectualism(1) | zestful(1) | sensitive(1) | determination(1) | dexterity(1) |
| hope(1) | visionary(1) | tenacity(1) | discipline(1) | depth(1) | audacity(1) |
| resourceful(1) | bonding(1) | passion(1) | preservation(1) | flair(1) | joyous(1) |
| reflective(1) | respect(1) | innovators(1) | heroic(1) | energetic(1) | kind-hearted(1) |
| remarkable(1) | identity(1) | zest(1) | peaceful(1) | minimalist(1) | optimistic(1) |
| enthusiastic(1) | bravery(1) | unyielding(1) | lively(1) | fervor(1) | epic(1) |
| adventurous(1) | genius(1) | serenity(1) | melodic(1) | celebration(1) | |

Table 21: Attribute words for Racial Bias Data Through Targeted Prompting

| | | Race Generation 3 | | | |
|---|---|---|---|---|---|
| resilience(9) | harmony(9) | innovative(8) | precision(7) | profound(7) | meticulous(7) |
| respect(7) | pioneering(6) | wisdom(5) | intricate(5) | innovation(5) | vibrant(5) |
| passion(4) | adaptability(4) | creativity(4) | unity(4) | blend(4) | unparalleled(4) |
| impeccable(4) | holistic(4) | sustainable(4) | artistry(3) | sustainability(3) | warmth(3) |
| resourcefulness(3) | courage(3) | acumen(3) | vitality(3) | wit(3) | functionality(3) |
| mindfulness(3) | forefront(3) | inclusivity(3) | audacious(3) | insights(3) | poetic(3) |
| serenity(3) | refreshing(3) | flair(3) | eloquence(2) | knowledge(2) | exploration(2) |
| hospitality(2) | introspection(2) | expertise(2) | tenacity(2) | legacy(2) | artistic(2) |
| freedom(2) | endurance(2) | love(2) | celebration(2) | strength(2) | essence(2) |
| harmonious(2) | enriching(2) | exceptional(2) | epitome(2) | boundless(2) | beacon(2) |
| genius(2) | dynamic(2) | pillars(2) | spiritual(2) | hope(2) | understanding(2) |
| mosaic(2) | strides(2) | marvels(2) | resonates(2) | philosophical(2) | reverence(2) |
| vivid(2) | astoundingly(2) | ethereal(2) | storytelling(2) | bonding(2) | inventive(2) |
| community(2) | spirituality(2) | adaptive(2) | joy(2) | compassion(2) | advocates(2) |
| modernity(2) | conservation(2) | contemporary(2) | guardianship(1) | visionary(1) | fluidity(1) |
| inquisitiveness(1) | innovations(1) | depth(1) | vastness(1) | tolerance(1) | agility(1) |
| magic(1) | vibrancy(1) | imagination(1) | solidarity(1) | oral(1) | enlightenment(1) |
| intricacies(1) | harmoniously(1) | grandeur(1) | bounty(1) | navigation(1) | emotions(1) |
| narratives(1) | history(1) | perspective(1) | depths(1) | heartbeat(1) | heritages(1) |
| entrepreneurial(1) | refined(1) | fresh(1) | adventure(1) | serene(1) | astuteness(1) |
| pivotal(1) | leading(1) | breakthrough(1) | critical(1) | vast(1) | wellspring(1) |
| cornerstone(1) | ingenuity(1) | elegance(1) | philosophy(1) | niche(1) | insight(1) |
| paramount(1) | brilliance(1) | leaders(1) | reflections(1) | lessons(1) | stewardship(1) |
| modernism(1) | instrumental(1) | windows(1) | relentless(1) | consciousness(1) | testament(1) |
| nexus(1) | symbols(1) | championing(1) | invaluable(1) | commentary(1) | templates(1) |
| reshaping(1) | indomitable(1) | merge(1) | pluralism(1) | seminal(1) | benchmarks(1) |
| agroecological(1) | reservoirs(1) | stories(1) | guardians(1) | resonant(1) | heartwarming(1) |
| steering(1) | canvas(1) | ecology(1) | morality(1) | smart(1) | agents(1) |
| illuminated(1) | icons(1) | interwoven(1) | commendable(1) | models(1) | enriched(1) |
| mesmerized(1) | exemplary(1) | echo(1) | genuine(1) | pacifistic(1) | introspective(1) |
| exploratory(1) | delightful(1) | eclectic(1) | groundbreaking(1) | futuristic(1) | zestful(1) |
| reflective(1) | inclusive(1) | joyful(1) | fascinating(1) | tranquil(1) | wistful(1) |
| whimsical(1) | rhythmic(1) | robust(1) | enigmatic(1) | indispensable(1) | contemplative(1) |
| altruistic(1) | intuitive(1) | detailed(1) | sagacious(1) | bold(1) | tenacious(1) |
| idyllic(1) | authentic(1) | monumental(1) | radiant(1) | cosmopolitan(1) | fearless(1) |
| penchant(1) | woven(1) | medicinal(1) | awe(1) | influence(1) | mesmerizing(1) |
| lyrical(1) | imbued(1) | existential(1) | captivating(1) | dedication(1) | minimalist(1) |
| timeless(1) | exquisite(1) | strikingly(1) | evocative(1) | exemplar(1) | remarkable(1) |
| introspectively(1) | amalgamation(1) | untouched(1) | heroism(1) | graceful(1) | richly(1) |
| pride(1) | successfully(1) | unique(1) | warmly(1) | enlightening(1) | refreshingly(1) |
| rooted(1) | profoundly(1) | touching(1) | enchanting(1) | impart(1) | compassionate(1) |
| imaginative(1) | revolutionizing(1) | sophisticated(1) | grace(1) | avant-garde(1) | audacity(1) |
| collaborative(1) | advancements(1) | caring(1) | adventurous(1) | craftsmanship(1) | strategic(1) |
| narrative(1) | enterprising(1) | maritime(1) | liberalism(1) | intellectual(1) | intrepid(1) |
| efficiency(1) | mutual(1) | engineering(1) | intensity(1) | aesthetic(1) | determination(1) |
| conservationist(1) | passionate(1) | perseverance(1) | finesse(1) | aesthetics(1) | vision(1) |
| melody(1) | bravery(1) | extraordinary(1) | spectrum(1) | diplomacy(1) | pacifism(1) |
| solace(1) | humor(1) | peace(1) | discipline(1) | justice(1) | democratic(1) |
| vegetarianism(1) | eco-friendly(1) | education(1) | humility(1) | mental health(1) | intercultural(1) |
| generosity(1) | renewable(1) | equality(1) | pedestrian-friendly(1) | collaboration(1) | support(1) |
| sportsmanship(1) | connections(1) | breakthroughs(1) | educational(1) | togetherness(1) | universal(1) |
| experiential(1) | kinship(1) | balance(1) | melodies(1) | interconnectedness(1) | well-being(1) |
| simplicity(1) | virtual reality(1) | diverse(1) | green(1) | interfaith(1) | protecting(1) |
| e-governance(1) | ancient(1) | linguistic(1) | | | |

Table 22: Attribute words for Religious Bias Data Through Targeted Prompting

| Religion Generation 1 | | | | |
|---|---|---|---|---|
| unity(7) | compassion(6) | peace(6) | integrates(6) | simplicity(6) | respect(6) |
| devotion(5) | music(4) | harmony(4) | health(4) | learning(4) | celebrate(4) |
| gratitude(4) | celebrates(4) | community(4) | charity(3) | equality(3) | brotherhood(3) |
| wisdom(3) | reverence(3) | clarity(3) | mystical(3) | joy(3) | art(3) |
| bonds(3) | divinity(3) | reflection(3) | journey(3) | history(3) | service(3) |
| mindfulness(3) | interplay(3) | vibrant(3) | balance(3) | insights(3) | redemption(3) |
| meditation(3) | synthesis(3) | heritage(3) | oneness(3) | bond(3) | artistic(3) |
| philosophical(2) | moral(2) | natural(2) | nature(2) | healing(2) | poetry(2) |
| rational(2) | musical(2) | craftsmanship(2) | dialogue(2) | cycles(2) | interpretations(2) |
| initiatives(2) | hospitality(2) | diverse(2) | theological(2) | well-being(2) | empowerment(2) |
| interconnectedness(2) | solace(2) | connection(2) | individualism(2) | enlightenment(2) | traditions(2) |
| recognition(2) | family(2) | mysteries(2) | symbols(2) | divine(2) | perseverance(2) |
| creator(2) | democratic(2) | tolerance(2) | purification(2) | insight(2) | energy(2) |
| compassionate(2) | knowledge(2) | innovation(2) | relationship(2) | mercy(2) | melodies(2) |
| blend(2) | renewal(2) | education(2) | symbolism(2) | culinary(2) | robotics(2) |
| architecture(2) | theatre(2) | engineering(2) | aerospace(2) | marine(2) | urban(2) |
| wildlife(2) | justice(2) | enlightening(1) | historical(1) | inspiring(1) | practical(1) |
| scientific(1) | personal(1) | spiritual(1) | mesmerizing(1) | legends(1) | cultural(1) |
| storytelling(1) | improvement(1) | individual(1) | benefit(1) | simplifies(1) | open(1) |
| governance(1) | dedication(1) | techniques(1) | genre(1) | ambiance(1) | architectural(1) |
| choirs(1) | celebrations(1) | principles(1) | resonate(1) | folklore(1) | thinking(1) |
| evidence(1) | hymns(1) | ethical(1) | narrates(1) | remedies(1) | life(1) |
| rhythmic(1) | preserves(1) | visualizations(1) | choral(1) | welcomes(1) | rite(1) |
| piety(1) | foundational(1) | depth(1) | profound(1) | earth(1) | align(1) |
| worship(1) | exploration(1) | rhythms(1) | magic(1) | sanctuary(1) | passion(1) |
| pacifism(1) | rites(1) | ancient(1) | vibrancy(1) | intimacy(1) | all-encompassing(1) |
| grace(1) | beacon(1) | harmonize(1) | humanitarian(1) | evangelism(1) | myths(1) |
| esoteric(1) | sovereignty(1) | nonviolence(1) | fellowship(1) | liturgical(1) | powerful(1) |
| solitude(1) | traditional(1) | alternative(1) | multiple(1) | inclusivity(1) | open-minded(1) |
| humanistic(1) | ancestral(1) | channel(1) | cultivate(1) | guidance(1) | connections(1) |
| bridges(1) | testament(1) | diversity(1) | progressive(1) | purity(1) | critical(1) |
| discipline(1) | generosity(1) | truth(1) | authentically(1) | poetic(1) | growth(1) |
| benevolence(1) | open-mindedness(1) | environment(1) | ethics(1) | worth(1) | honor(1) |
| scholarship(1) | reason(1) | ritual(1) | mythological(1) | perspective(1) | practices(1) |
| bridge(1) | mysticism(1) | self-empowerment(1) | celebration(1) | embraces(1) | enlightened(1) |
| ministry(1) | misconceptions(1) | performance(1) | connect(1) | colors(1) | accordance(1) |
| hope(1) | interwoven(1) | cyclical(1) | yearning(1) | sustainability(1) | technology(1) |
| athletics(1) | mathematician(1) | ecology(1) | physicist(1) | entrepreneurship(1) | linguistic(1) |
| leadership(1) | software(1) | astronomy(1) | fashion(1) | finance(1) | biology(1) |
| genetic(1) | renewable(1) | intelligence(1) | dance(1) | philanthropy(1) | diplomat(1) |
| animation(1) | data(1) | environmental(1) | graphic(1) | medicinal(1) | virtual(1) |
| nanotechnology(1) | coding(1) | chemical(1) | farming(1) | astrophysicist(1) | biotechnology(1) |
| neurosciences(1) | computational(1) | futuristic(1) | digital(1) | geology(1) | organic(1) |
| literature(1) | gaming(1) | quantum(1) | photography(1) | abstract(1) | climatologist(1) |
| neurology(1) | fiction(1) | bioinformatics(1) | genomics(1) | pottery(1) | journalist(1) |
| analytics(1) | cybersecurity(1) | linguistics(1) | evolutionary(1) | forensic(1) | agricultural(1) |
| software engineer(1) | quantum computing(1) | landscape painting(1) | aerodynamics(1) | environmental law(1) | animation and design(1) |
| particle physics(1) | cryptography(1) | molecular biology(1) | ethnomusicology(1) | digital marketing(1) | sustainable energy solutions(1) |
| immersive technology(1) | documentary filmmaking(1) | neurosurgical advancements(1) | social entrepreneurship(1) | urban forestry(1) | data visualization(1) |
| charitable(1) | non-violence(1) | helping(1) | kindness(1) | intellectual(1) | selfless(1) |
| philosophy(1) | thoughts(1) | valor(1) | integrity(1) | righteous(1) | cherishes(1) |
| connectivity(1) | feminine(1) | heal(1) | emotional(1) | tranquility(1) | self-awareness(1) |
| disciplined(1) | love(1) | interpretation(1) | peaceful(1) | histories(1) | questioning(1) |
| lack(1) | variety(1) | single(1) | forces(1) | pacifist(1) | dedicated(1) |
| self-improvement(1) | soulful(1) | outreach(1) | contemplation(1) | journeys(1) | milestones(1) |
| harmonious(1) | | | | | |

Table 23: Attribute words for Religious Bias Data Through Targeted Prompting

| Religion Generation 2 | | | | | |
|---|---|---|---|---|---|
| mindfulness(7) | ethical(6) | unity(5) | philosophical(5) | compassion(4) | ecological(4) |
| wisdom(5) | harmony(4) | historical(5) | governance(4) | education(4) | poetry(4) |
| service(4) | knowledge(3) | literature(3) | cultural(4) | nature(3) | music(3) |
| peace(3) | gratitude(3) | humanitarian(4) | arts(3) | environmental(3) | insights(3) |
| charity(3) | dance(3) | musical(4) | balance(3) | meditation(3) | interpretations(3) |
| ancient(3) | worship(3) | science(2) | astronomy(2) | humility(2) | artists(2) |
| resilience(2) | conservation(2) | justice(2) | linguistic(2) | psychological(2) | craftsmanship(2) |
| loyalty(2) | preservation(2) | psychology(2) | pacifist(2) | theology(2) | diplomacy(2) |
| sustainability(2) | rebirth(2) | wellness(2) | engagement(2) | literacy(2) | bonds(2) |
| poetic(3) | architectural(3) | business(2) | reflection(2) | welfare(2) | leadership(2) |
| non-violence(2) | scholarship(2) | community(2) | learning(2) | family(2) | cycles(2) |
| symbolism(2) | simple(3) | teachings(2) | liturgical(2) | joyous(3) | harmonizing(2) |
| integrates(2) | distinct(2) | innovative(1) | philanthropic(1) | poets(1) | reverence(1) |
| selfless(2) | mathematical(1) | charitable(2) | scientists(1) | philosophy(1) | physics(1) |
| socio-political(1) | supportive(1) | herbal(1) | biodiversity(1) | empowerment(1) | folktales(1) |
| vibrant(1) | art(1) | mental(1) | societal(1) | growth(1) | political(1) |
| dialogue(1) | joy(1) | preserved(1) | perspectives(1) | cohesion(1) | introspection(1) |
| inquiry(1) | existentialism(1) | enlightenment(1) | wonder(1) | amalgamation(1) | debates(1) |
| aesthetics(1) | tolerance(1) | inclusivity(1) | autonomy(1) | simplicity(2) | translation(1) |
| sociological(1) | exchange(1) | beauty(1) | kindness(1) | scholars(1) | technological(1) |
| advocates(1) | modern(1) | quantum(1) | jazz(1) | interfaith(1) | progressive(1) |
| development(1) | organic(1) | philanthropist(1) | artistry(1) | activism(1) | astronomers(1) |
| classical(1) | organizational(1) | sanctuary(1) | sports(1) | stem(1) | negotiation(1) |
| holistic(1) | academic(1) | healing(1) | plantation(1) | archaeological(1) | botanical(1) |
| fashion(1) | storytelling(1) | vocational(1) | relief(1) | culinary(1) | preserving(2) |
| understanding(1) | humanities(1) | environmentalism(1) | photographers(1) | bonding(1) | hospitality(1) |
| rationalism(1) | therapeutic(1) | medicine(1) | outreach(1) | genealogy(1) | moral(2) |
| sustainable(1) | resolution(1) | cinema(1) | sciences(1) | cosmos(1) | reconciliation(1) |
| astronomical(1) | environmentalists(1) | entrepreneurship(1) | philanthropy(1) | intellectualism(1) | ethics(1) |
| equality(1) | healthcare(1) | thoughts(1) | cooperation(1) | perseverance(1) | pride(1) |
| interconnectedness(1) | diversity(1) | psyche(1) | aid(1) | land(1) | baptism(1) |
| sung(1) | well-being(2) | self-discovery(1) | chanting(2) | truths(1) | purification(1) |
| peaceful(2) | esoteric(1) | early(1) | myths(1) | open-minded(1) | reason(1) |
| universe(1) | diverse(2) | unifying(2) | interplay(1) | synthesis(1) | one(1) |
| mercy(1) | inner(1) | grace(1) | bridge(1) | prioritize(1) | health(1) |
| evangelism(1) | self-improvement(1) | services(1) | texts(1) | renewal(1) | milestones(1) |
| sanctity(1) | integrity(1) | harmonious(2) | betterment(1) | honor(1) | triumph(1) |
| inspiring(1) | intricate(1) | transcendent(1) | guiding(1) | insightful(1) | hopeful(1) |
| community-driven(1) | solemn(1) | balancing(1) | responsibility(1) | creation(1) | resilient(1) |
| loving(1) | ancestral(1) | life-affirming(1) | reverent(1) | seasonal(1) | fertility(1) |
| health-maintaining(1) | combining(1) | rhythmic(1) | oral(1) | nature-bound(1) | detailed(1) |
| meditative(1) | self-explorative(1) | empowering(1) | clarity(1) | original(1) | quick(1) |
| introspective(1) | theological(1) | dialogic(1) | mystical(2) | alternative(1) | kinship(1) |
| festive(1) | folkloric(1) | open(1) | questioning(1) | evidence-based(1) | creator(2) |
| universal(1) | all-encompassing(1) | opposing(1) | unified(1) | redemptive(1) | silent(1) |
| advocating(1) | sovereign(1) | graceful(1) | choral(1) | democratic(1) | traditional(1) |
| purifying(1) | evangelistic(1) | clearing(1) | soulful(1) | communal(1) | testament(1) |
| social(1) | solitudinous(1) | shared(1) | comforting(1) | interconnected(1) | profound(1) |
| guideline(1) | fostering(1) | connecting(1) | celebrate(1) | journeying(1) | homage(1) |
| blending(1) | integrating(1) | delving(1) | challenging(1) | solace(1) | context(1) |
| personal(1) | monotheistic(1) | hymns(1) | origin(1) | morality(1) | laws(1) |
| eternal(1) | history(1) | mix(1) | spirit(1) | witchcraft(1) | communicating(1) |
| deities(1) | syncretic(1) | self-help(1) | focuses(1) | oldest(1) | bodhisattva(1) |
| mantra(1) | strict(1) | persecution(1) | joyful(1) | mesopotamian(1) | liberal(1) |
| skepticism(1) | asserts(1) | divine(1) | multiple(1) | single(1) | dichotomy(1) |
| combines(1) | oneness(1) | salvation(1) | light(1) | name(1) | predestination(1) |
| emerged(1) | retains(1) | decentralized(1) | initiation(1) | sabbath(1) | writings(1) |
| sacred(1) | traditions(1) | secluded(1) | journey(1) | expressing(1) | challenge(1) |
| chant(1) | guidance(1) | recognizes(1) | champions(1) | | |

Table 24: Attribute words for Religious Bias Data Through Targeted Prompting

| Religion Generation 3 | | | | | |
|---|---|---|---|---|---|
| balance(9) | unity(7) | mystical(8) | harmony(7) | mindfulness(6) | love(5) |
| community(7) | compassion(5) | equality(5) | simplicity(6) | healing(4) | joy(5) |
| nature(5) | salvation(4) | peace(5) | divinity(4) | ethical(3) | wisdom(4) |
| respect(4) | integration(3) | meditation(3) | gratitude(4) | intricate(3) | liturgical(4) |
| esoteric(4) | pacifism(3) | grace(3) | music(3) | commitment(3) | development(3) |
| insights(4) | architectural(3) | solace(4) | ancient(5) | integrates(3) | democratic(3) |
| knowledge(3) | dialogue(2) | good(2) | cultural(3) | family(3) | integrity(3) |
| learning(3) | heritage(2) | cyclical(2) | beauty(2) | purifying(2) | individualism(3) |
| transformative(3) | poetic(3) | kinship(2) | poetry(2) | secular(2) | blends(2) |
| engagement(2) | transformation(2) | ethics(3) | charity(3) | non-violence(3) | service(2) |
| spiritual(2) | blend(2) | diverse(3) | singular(2) | oneness(3) | rebirth(3) |
| health(2) | evangelism(2) | central(2) | songs(2) | justice(3) | perspectives(2) |
| honor(2) | interpretation(2) | combines(2) | celebrate(2) | history(2) | multiple(2) |
| journey(2) | bridge(2) | peaceful(1) | science(1) | sustainability(1) | selfless(1) |
| scholarship(1) | charitable(1) | pioneering(1) | education(1) | humanitarian(1) | healthcare(1) |
| art(1) | environment(1) | community-building(1) | resilience(2) | synthesis(1) | preservation(1) |
| blending(1) | togetherness(2) | preserve(1) | self-awareness(2) | responsibility(1) | benefit(1) |
| jurisprudential(1) | interfaith(1) | illumination(1) | exploration(2) | reason(1) | diversity(2) |
| interplay(2) | dignity(1) | sovereignty(1) | craftsmanship(1) | renewal(1) | well-being(1) |
| study(1) | devotion(1) | exchange(1) | artistic(1) | musical(1) | contemplation(1) |
| connection(2) | profound(1) | interconnectedness(1) | universal(1) | philosophy(1) | fellowship(1) |
| continuity(1) | conduct(1) | self-respect(1) | ancestors(1) | rhythms(1) | ancestral(1) |
| purity(1) | truthfulness(1) | consciousness(1) | happiness(1) | original(1) | enlightenment(2) |
| influential(1) | moderation(1) | egyptian(1) | reincarnation(1) | festivals(1) | integrate(1) |
| explore(1) | non-interventionist(1) | pantheon(1) | acceptance(1) | simple(1) | theological(2) |
| dating(1) | traditional(1) | teachings(1) | prayer(1) | significance(1) | discipline(1) |
| structure(1) | align(1) | joyful(1) | communion(1) | predestination(1) | participation(1) |
| freedom(1) | foundational(1) | reverence(2) | support(1) | transitions(1) | humility(2) |
| kindness(2) | perseverance(1) | brotherhood(1) | purpose(1) | faith(2) | connections(1) |
| revere(1) | vibrant(1) | meditative(1) | thinking(2) | belonging(2) | sacredness(1) |
| spirit(2) | expressions(1) | growth(2) | silence(1) | reaffirm(1) | symbolism(1) |
| righteousness(2) | forgiveness(1) | collective(1) | hymns(2) | sanctuary(1) | improvement(1) |
| culture(2) | modernity(2) | foundations(1) | humanism(1) | welcomes(1) | believes(1) |
| manuscripts(1) | holistic(1) | introspection(1) | thought(1) | universe(1) | tapestry(1) |
| sentient(1) | joyous(1) | clarity(1) | champion(1) | syncretism(1) | loyalty(2) |
| inclusivity(1) | rectitude(1) | alternative(1) | cycles(2) | enlightening(1) | scholarly(1) |
| patience(1) | truth(1) | oldest(1) | dedication(1) | inspiration(1) | tranquil(1) |
| serenity(1) | discovery(1) | hubs(1) | iconography(1) | quest(1) | inquiry(1) |
| distant(1) | divine(1) | supreme(1) | rooted(1) | vast(1) | range(1) |
| phases(1) | traditions(1) | largest(1) | humble(1) | roots(1) | tantra(1) |
| preserved(1) | worship(1) | misunderstood(1) | african(1) | perspective(1) | spirits(1) |
| lotus(1) | spread(1) | betterment(1) | bodhisattva(1) | moral(1) | creator(1) |
| goddess(1) | guidance(1) | self-discipline(1) | beacon(1) | earliest(1) | jurisprudence(1) |
| eternal(1) | devotee's(1) | honesty(1) | hospitality(1) | relationship(1) | conservation(1) |
| creation(1) | philosophical(1) | guidelines(1) | families(1) | dances(1) | seasons(1) |
| nature-oriented(1) | storytelling(1) | elevate(1) | autonomy(1) | chanting(1) | monastic(1) |
| symbolic(1) | interpretations(1) | deeper(1) | mystic(1) | unknown(1) | rational(1) |
| detached(1) | incorporated(1) | open-minded(1) | structured(1) | wellness(1) | tools(1) |
| narrate(1) | centers(1) | serene(1) | familial(1) | depths(1) | |

Table 25: Attribute words for General Prompting Data

| General Generation 1 | | | | | |
|---|---|---|---|---|---|
| poetry(10) | ballet(6) | chess(6) | astrophysics(6) | literature(5) | opera(5) |
| astronomy(5) | art(4) | farming(4) | salsa(4) | environmental(4) | calligraphy(4) |
| robotics(4) | pottery(4) | ballroom(4) | coding(4) | dance(4) | physics(4) |
| yoga(3) | meditation(3) | mathematics(3) | archaeology(3) | programming(3) | dancer(3) |
| theater(3) | wildlife(3) | comedy(3) | marine(3) | quantum(3) | skiing(3) |
| fashion(3) | jazz(3) | sustainable(3) | novels(3) | entomology(3) | strongest(2) |
| struggled(2) | quantum physics(2) | violin(2) | weightlifter(2) | psychology(2) | martial arts(2) |
| historian(2) | economics(2) | birdwatching(2) | vegan(2) | marine biology(2) | pianist(2) |
| tech(2) | jewelry(2) | astronomer(2) | sports(2) | gourmet(2) | renaissance(2) |
| volunteering(2) | tech-savvy(2) | mechanic(2) | baking(2) | financial(2) | wilderness(2) |
| garden(2) | gaming(2) | organic(2) | climbing(2) | biology(2) | mechanical(2) |
| culinary(2) | historical(2) | archaeological(2) | ornithology(2) | chemistry(2) | anthropology(2) |
| swimmer(2) | volunteered(2) | mountaineer(2) | neuroscience(2) | rock climber(2) | bird(2) |
| botany(2) | mechanics(2) | piano(2) | pastry(2) | sculpture(2) | symphonies(2) |
| origami(2) | technology(1) | sensitivity(1) | photographer(1) | timeless(1) | snowboarding(1) |
| anonymously(1) | books(1) | fastest(1) | party(1) | eloquent(1) | outdoor(1) |
| volunteer(1) | children's hospitals(1) | mountain climber(1) | progressive(1) | rock star(1) | flexibility(1) |
| gentle(1) | genius(1) | romantic(1) | environmentalist(1) | paintings(1) | simple(1) |
| classical literature(1) | biologist(1) | rescue(1) | florist(1) | mental health(1) | cupcakes(1) |
| sunniest(1) | space exploration(1) | composed(1) | marathons(1) | linguistics(1) | harp(1) |
| paint(1) | floral(1) | basketball(1) | skateboarder(1) | kindergarten(1) | kendo(1) |
| ranger(1) | romance(1) | decorator(1) | dancing(1) | dj(1) | neuroscientist(1) |
| graffiti(1) | musician(1) | comedian(1) | scuba(1) | cooking(1) | creativity(1) |
| musical(1) | mathematical(1) | embroidery(1) | physical(1) | digital(1) | scientific(1) |
| botanists(1) | designing(1) | breakdancing(1) | ornithological(1) | handicrafts(1) | expeditions(1) |
| history(1) | racing(1) | butterflies(1) | energy(1) | fantasy(1) | aerospace(1) |
| technologies(1) | animation(1) | documentaries(1) | conservation(1) | architectural(1) | sculptors(1) |
| planning(1) | martial(1) | design(1) | philosophy(1) | neural(1) | orchestras(1) |
| biochemistry(1) | aerodynamics(1) | sociology(1) | climate(1) | microbiological(1) | geology(1) |
| game(1) | musicians(1) | acrobatic(1) | pianists(1) | nanotechnology(1) | compassionate(1) |
| work ethic(1) | zoos(1) | gender equality(1) | kindest(1) | ballet dancer(1) | diligent(1) |
| global politics(1) | lgbtq+ rights(1) | gun(1) | wisdom(1) | leader(1) | humor(1) |
| surf(1) | desert ecology(1) | traditional cultures(1) | athletic(1) | public speaking(1) | beaches(1) |
| cinema(1) | renaissance art(1) | pasta(1) | rehabilitating(1) | vision(1) | community service(1) |
| forest conservation(1) | italian cuisine(1) | ancient history(1) | alpine flora(1) | classical music(1) | underwater archaeology(1) |
| sustainable living(1) | wildlife conservation(1) | particle physics(1) | diver(1) | vodka(1) | animal rights(1) |
| tea(1) | botanist(1) | mathematician(1) | car(1) | potter(1) | civil rights(1) |
| mathematical theorem(1) | bullfighting(1) | gourmet chef(1) | non-violence(1) | quantum physicist(1) | african tribal music(1) |
| butterfly collection(1) | rocket scientist(1) | workers' rights(1) | rally driver(1) | snails(1) | cardiovascular surgeon(1) |
| skydiver(1) | cellist(1) | marine engineer(1) | nuclear physics(1) | author(1) | software developer(1) |
| spicy food(1) | maestro(1) | art historian(1) | quantum mechanics(1) | linguistic(1) | nuclear chemist(1) |
| cold(1) | artificial intelligence(1) | wildlife photography(1) | hacker(1) | knitting(1) | zoo(1) |
| karate(1) | book(1) | nurturing(1) | peace(1) | butchers(1) | painting(1) |
| lessons(1) | global(1) | photographic(1) | blacksmith(1) | rugby(1) | gratitude(1) |
| kickboxing(1) | compassion(1) | wines(1) | struggles(1) | avant-garde(1) | ride(1) |
| astrophysicist(1) | renewable(1) | flamenco(1) | abstract(1) | impressionist(1) | rock(1) |
| urban(1) | molecular(1) | classical(1) | volleyball(1) | greek(1) | mindful(1) |
| cello(1) | rural(1) | circus(1) | woodworking(1) | surfing(1) | ai(1) |
| permaculture(1) | particle(1) | beach(1) | hockey(1) | deep-sea(1) | desert(1) |
| neurobiology(1) | jiu-jitsu(1) | bagpipes(1) | rodeo(1) | rose(1) | tattoo(1) |
| knit(1) | motorcycles(1) | active(1) | watercolor(1) | stargazing(1) | authored(1) |
| flutist(1) | participate(1) | sunny(1) | poet(1) | ph.d.(1) | skydiving(1) |
| sci-fi(1) | solve(1) | beekeeping(1) | gardening(1) | bonsai(1) | virtual reality(1) |
| fluent(1) | gamer(1) | digital art(1) | race car(1) | archery(1) | philosopher(1) |
| archeology(1) | tango(1) | metal(1) | rescuing(1) | guitar(1) | acrobatics(1) |
| surfer(1) | skater(1) | storybook(1) | capoeira(1) | boxing(1) | motorcycle(1) |
| fencing(1) | esports(1) | engineering(1) | breakdance(1) | saxophonist(1) | mural(1) |
| falconry(1) | tennis(1) | didgeridoo(1) | punk(1) | scuba diving(1) | |

Table 26: Attribute words for General Prompting Data

| General Generation 2 | | | | | |
|---|---|---|---|---|---|
| astrophysics(12) | poetry(8) | ballet(7) | coding(6) | chess(6) | literature(6) |
| conservation(5) | novels(5) | innovative(4) | quantum physics(4) | politics(4) | philosophy(4) |
| opera(4) | violin(4) | aerospace(4) | shakespeare(3) | tech-savvy(3) | maestro(3) |
| peace(3) | meditation(3) | pottery(3) | neuroscience(3) | vegan(3) | ornithology(3) |
| historian(3) | salsa(3) | sculptor(3) | mountaineer(3) | physics(3) | history(3) |
| biologist(3) | pianist(3) | research(3) | calculus(2) | classical(2) | mathematician(2) |
| culinary(2) | astronomy(2) | leadership(2) | entrepreneurial(2) | gardening(2) | farming(2) |
| martial artist(2) | yoga(2) | mathematical(2) | adventure(2) | animal rights(2) | nuclear physics(2) |
| comedy(2) | archeology(2) | author(2) | mental health(2) | mindfulness(2) | quantum mechanics(2) |
| astrophotography(2) | sociology(2) | ballroom(2) | harp(2) | poets(2) | ph.d.(2) |
| wisdom(2) | novel(2) | art(2) | sunny(2) | technologies(2) | academic(2) |
| physicist(2) | biology(2) | gourmet(2) | ornithologist(2) | scientist(2) | judo(2) |
| mechanics(2) | archaeology(2) | computing(2) | playwright(2) | chemistry(2) | garden(2) |
| paintings(2) | sustainable(2) | archaeological(2) | robotics(2) | languages(2) | martial arts(2) |
| architecture(2) | violinist(2) | leaders(2) | scientific(2) | tech(2) | botanical(2) |
| scholars(2) | marine biologist(2) | classical music(2) | space exploration(2) | digital(2) | mathematics(3) |
| molecular biology(2) | quantum computing(2) | economics(2) | nurturing(1) | adapt(1) | gentle(1) |
| strength(1) | work ethic(1) | technology(1) | rapport(1) | scholar(1) | literary(1) |
| community service(1) | botanist(1) | renaissance(2) | classical literature(1) | optimistic(1) | acumen(1) |
| sports enthusiast(1) | gadgets(1) | dance(1) | public speaker(1) | ancient crafts(1) | jazz(1) |
| virtual reality(1) | stamp collection(1) | astronomer(1) | multilingual(1) | volunteered(1) | women's rights(1) |
| painter(1) | yoga instructor(1) | theater(1) | environmental science(1) | marathons(1) | homeless(1) |
| tutored(1) | karate(1) | grassroots(1) | swimmer(1) | documentary(1) | magician(1) |
| tango(1) | cookbooks(1) | poetry slams(1) | digital animation(1) | roller derby(1) | jazz prodigy(1) |
| calligraphy(1) | puppeteer(1) | created(1) | mathematicians(1) | drivers(1) | teach(1) |
| humble(1) | volunteering(1) | martial(1) | party(1) | philosopher(1) | renewable(1) |
| patents(1) | singing(1) | fluent(1) | conservationists(1) | understand(1) | wizard(1) |
| compassionate(1) | basketball(1) | botany(2) | activists(1) | fashion(1) | proust(1) |
| biochemist(1) | books(1) | vegetables(1) | wine(1) | archery(1) | poet(1) |
| cooking(1) | podcast(1) | greek(1) | professor(1) | painting(1) | civilizations(1) |
| bestselling(1) | prodigy(1) | dancing(1) | stories(1) | comedian(1) | equestrian(1) |
| filmmaker(1) | entomology(1) | charity(1) | coded(1) | entrepreneurship(1) | sitar(1) |
| cuisine(1) | crochet(1) | uplift(1) | trading(1) | scholarships(1) | restoration(1) |
| debates(1) | programming(1) | veganism(1) | beekeeping(1) | diplomacy(1) | cookbook(1) |
| healing(1) | paleontology(1) | driver(1) | marketing(1) | ocean(1) | welfare(1) |
| resolution(1) | explorer(1) | inventions(1) | guitarist(1) | journals(1) | rescue(1) |
| couture(1) | culture(1) | composition(1) | cello(1) | fencing(1) | nano-technology(1) |
| flute(1) | neurobiology(1) | artwork(1) | cyber-security(1) | engineering(1) | intelligence(1) |
| actress(1) | animation(1) | skydiver(1) | photography(1) | saxophone(1) | clarinet(1) |
| mythology(1) | musician(1) | courageous(1) | groundbreaking(1) | caregivers(1) | innovation(1) |
| contribute(1) | respect(1) | beautifully(1) | pioneering(1) | captivating(1) | understanding(1) |
| contributions(1) | fluently(1) | delicate(1) | enriched(1) | well-being(1) | nature(1) |
| academically(1) | service(1) | adopting(1) | excel(1) | innovations(1) | insights(1) |
| analytical(1) | technological(1) | ai technology(1) | family time(1) | adventurous(1) | arts(1) |
| trailblazers(1) | competitive(1) | cosmology(1) | stem(1) | adventurers(1) | dancers(1) |
| activism(1) | mountaineering(1) | emotional support(1) | fine art(1) | theoretical physics(1) | dramatic arts(1) |
| breakthrough(1) | astronomical(1) | authors(1) | sculptors(1) | ballroom dancing(1) | particle physics(1) |
| environmental sciences(1) | oceanography(1) | marine biologists(1) | football(1) | extreme sports(1) | algorithms(1) |
| donate(1) | environmental(1) | social work(1) | telecommunication(1) | baking(1) | cinema(1) |
| astrophysicist(1) | urban planning(1) | cosmos(1) | ancient civilizations(1) | aerodynamics(1) | filmmaking(1) |
| app development(1) | folklore(1) | nuclear physicist(1) | philosophical(1) | microbiology(1) | music(1) |
| astrophysical(1) | environmentalist(1) | digital graphics(1) | computer programming(1) | reptile handling(1) | jazz history(1) |
| renewable energy(1) | plant biology(1) | african dances(1) | economic theories(1) | renaissance art(1) | engineer(1) |
| psychology(1) | wildlife photographer(1) | biochemistry(1) | anthropology(1) | botanical research(1) | fashion designer(1) |
| aerospace engineering(1) | weightlifting(1) | symphonic(1) | | | |

