# OpenReview forum: "ChatGPT Based Data Augmentation for Improved Parameter-Efficient Debiasing of LLMs"
_colmweb.org/COLM/2024/Conference — COLM_

### Official Review · Reviewer_n66S · 2024-05-11

**Rating:** 6
**Confidence:** 4
**Ethics Flag:** 1

**Summary:**

This paper proposes a novel data augmentation framework for the purpose of social bias mitigation by leveraging the generative power of ChatGPT. The motivation for this paper is reasonable and well supported. Furthermore, improving the quality of a debiasing corpus is a straightforward and effective solution. However, the experimental results of BERT are not very strong.

**Reasons To Accept:**

- The paper is well presented.
- This paper focuses on one of the most significant issues of social bias mitigation.
- The proposed framework can be reproduced easily and more than one social bias has been considered.

**Reasons To Reject:**

- One possible issue is the generated sentences might still be biased or toxic, even though ChatGPT is prompted not to be. Therefore, the paper could be improved by including validation results about the harmfulness of those generations.
- Given that the LoRA or adaptor tuning is a computationally cheap solution for debiasing, this means that we can anticipate that the adaptation layer can debias the learned internal knowledge by referring to the debiasing corpus. How many adaptation layers do we need for this purpose? What is the gap between a solution that debiases the whole network and the solution with an adaptation layer? Debiasing a small BERT model would be expensive enough, though it would strengthen the paper to have other references or tiny experiments for this. However, if the authors have hardware limitations, please ignore this.
- I am not that surprised about the potential ineffectiveness of the generated sentences for mitigating social bias from smaller LLMs, such as BERT or GPT-neo. Borrowing from the research outcomes of theoretical understanding towards in-context learning [1], if we consider that pretraining/in-context learning is to learn/locate a class of function w.r.t. the training corpus, the learned function classes for ChatGPT and BERT should be very different because of the model scales (or maybe other additional factors). I would appreciate it if our authors can suggest some solutions in this respect. A relevant case might be self-alignment [2] or self-instruction [3].

[1] Mao, Haitao, et al. "A Data Generation Perspective to the Mechanism of In-Context Learning." arXiv preprint arXiv:2402.02212 (2024).
[2] Sun, Zhiqing, et al. "Principle-driven self-alignment of language models from scratch with minimal human supervision." Advances in Neural Information Processing Systems 36 (2024).
[3] Wang, Yizhong, et al. "Self-Instruct: Aligning Language Models with Self-Generated Instructions." Proceedings of the 61st Annual Meeting of the Association for Computational Linguistics (Volume 1: Long Papers). 2023.

---

> ### Author Rebuttal · Authors · 2024-05-29
>
> Dear Reviewer,
>
> Thank you for your detailed assessment and recognition of our efforts. We would like to address the specific points mentioned:
>
> 1. **Potential Introduction of Biases:**
>    We acknowledge that even with anti-stereotype prompts, ChatGPT can introduce nuanced biases or implicit toxicity, which may remain undetected despite our careful inspection (see Figure 4, Table 1, and Appendices). However, we highlight the following:
>    - **Effectiveness:** The same synthetic dataset (synthetic-general) is effective at debiasing across several bias categories in different datasets, suggesting it doesn’t introduce new biases while mitigating others.
>    - **Comparison with Real-World Data:** Real-world data collection is challenging and largely dependent on human input, which can vary in reliability and bias. In contrast, our synthetic data, generated with a thoughtfully designed prompting strategy, is more efficient, consistent, and offers a higher degree of control.
>    - **Area for Improvement:** A potential additional check that might be added is another expert LLM to check for toxicity, using models like ToxicBERT or Self-Detoxification.
>
> 2. **Experimental Setup Clarifications:**
>    - **Limited Resources and Adapter Layers**: Our experiments on Google Colab use default settings with an adapter layer after each transformer layer (12 layers for bert-base-uncased). We will include these details in the paper.
>    - **Full Finetuning:** Full-finetuning/retraining is computationally expensive and may lead to overfitting given our dataset is small. Additionally, the work by Xie and Lukasiewicz, which we compare with, shows that parameter-efficient methods are better. We will discuss this in the paper.
>
> 3. **Distribution Differences and Tailored Data Generation:**
>    - **Distribution Differences:** Since distributions differ between smaller and larger models, as well as between GPT and non-GPT families, it’s essential to tailor data generation. Our loss-guided prompting strategy (described in Section 3) addresses this. We generate data normally, calculate the loss using model M (the model being debiased), and use lower-loss data as few-shot examples to generate more in-distribution data, reducing the distribution gap.
>    - **Alternative Methods:** We will discuss self-instruct and self-aligned methods as other possible alternatives.
>
> We hope our explanations address your concerns. Please let us know if there are any further questions or suggestions. Thank you!

---

> > ### Author Response · Authors · 2024-06-06
> >
> > We once again appreciate all the comments and we hope our rebuttal helped clarify and address the remaining concerns. Please let us know if we can further clarify anything. We would also like to point to the additional analysis of Toxicity, which we performed in response to the latest comments from Reviewer fjaf.

---

### Official Review · Reviewer_P4Tc · 2024-05-11

**Rating:** 7
**Confidence:** 4
**Ethics Flag:** 1

**Summary:**

The paper introduces an interesting approach to debiasing large language models (LLMs) using synthetic data generated via ChatGPT, employing targeted and general prompting strategies alongside an auxiliary loss-guided prompting method. By adapting adapter tuning, this technique efficiently mitigates biases such as racial, gender, and religious prejudices with minimal degradation of language capabilities. Comparative evaluations against traditional datasets demonstrate that this method not only competes favorably but often exceeds existing debiasing strategies in effectiveness while substantially reducing computational costs and training time. The findings suggest that synthetic data can significantly advance the fairness of LLMs.

**Questions To Authors:**

See weakness

**Reasons To Accept:**

1. Debiasing Methodology: Introduces a novel approach that uses ChatGPT to generate high-quality synthetic data, enhancing the robustness of debiasing in a parameter-efficient manner.
2. Dual Prompting Strategies: Proposes two synthetic data generation methods: targeted prompting for specific bias mitigation requiring prior bias knowledge and general prompting for broader bias mitigation without needing prior knowledge, albeit with slightly less effectiveness.
3. Loss-Guided Prompting Enhancement: Experiments with a new variant of targeted prompting called loss-guided prompting show promising results specifically for improving the debiasing performance of the BERT model.

**Reasons To Reject:**

1. While with a slightly different motivation, the method of this work seems to be similar to other attributed prompting-based debiasing methods like Attrprompt [1]. [1] Yu, Yue, et al. "Large language model as attributed training data generator: A tale of diversity and bias." Advances in Neural Information Processing Systems 36 (2024).
2. Existing related works in Section 2.2 seem to focus on debiasing methods before LLMs. Could you provide a comprehensive review of debiasing methods for LLMs as baselines?

---

> ### Author Rebuttal · Authors · 2024-05-29
>
> Dear Reviewer,
>
> We sincerely appreciate your thoughtful review and the recognition of our paper’s contributions. We would like to respond to the concerns raised.
>
>
> - **Regarding the first point**, while synthetic data generation and augmentation are not novel concepts, our approach to utilizing them for effective debiasing is innovative in this context. The method introduced in the work by Yu et al. [1] focuses on generating more diverse, high-quality, and unbiased data by controlling various attributes such as length, style, and location. Their approach prompts the model to generate data considering multiple dimensions (e.g., [Generate A, consider Dimension1, Dimension2, Dimension3, etc.]). In contrast, our method specifically leverages the model’s knowledge of social bias and social groups to generate anti-stereotyped data, aiming to counteract bias. Our attributes refer to names and entities within social groups, and our [Sentence, Target, Attribute] format is designed to support subsequent pipelines for effective debiasing via parameter-efficient fine-tuning. In summary, while Yu et al. [1]’s method uses prompting strategies to create diverse and unbiased synthetic datasets, our method actively leverages ChatGPT’s knowledge of social bias to generate targeted anti-stereotyped data for debiasing. We differ in goals and approaches.
>
> - **Regarding the second point**, we apologize for any confusion. Current popular debiasing methods for LLMs generally involve 1)  fine-tuning or retraining LLMs on specific datasets. 2) applying post-hoc mathematical operations on the frozen representations of a LM.  The first two paragraphs in Section 2.2 of our paper introduce various attempts to build high-quality datasets for fine-tuning or retraining models. In the last paragraph, we mentioned the work by Xie and Lukasiewicz, which provides a comprehensive evaluation of different major parameter-efficient fine-tuning and model retraining methods for debiasing purposes. Their work also includes post-hoc methos like SentenceDebias (Liang et al., 2020) and SelfDebias (Schick et al., 2021). This work serves as our direct comparison and baselines.
>
>
> Thank you once again for recognizing our potential and providing valuable suggestions. We hope these clarifications address your concerns, and we are happy to answer any further questions and incorporate these clarificaitons into the paper.

---

> > ### Author Response · Authors · 2024-06-06
> >
> > We once again appreciate all the comments and we hope our rebuttal helped clarify and address the remaining concerns. Please let us know if we can further clarify anything.

---

### Official Review · Reviewer_fjaf · 2024-05-12

**Rating:** 6
**Confidence:** 3
**Ethics Flag:** 1

**Summary:**

This paper introduces a novel approach that utilizes ChatGPT to generate synthetic training data, with the aim of improving the debiasing process of Large Language Models (LLMs). The authors propose two strategies: Targeted Prompting, which effectively addresses known biases but requires prior specification of the bias in question, and General Prompting, which offers debiasing across various categories albeit with slightly less effectiveness.

The paper leverages resource-efficient LLM debiasing using adapter tuning and compares the effectiveness of the synthetic data generated by the approach to existing debiasing datasets. The results reveal several key findings: Firstly, ChatGPT can efficiently produce high-quality training data for debiasing other LLMs. Secondly, the data generated via their approach outperforms existing datasets in terms of debiasing performance while also preserving the internal knowledge of a pre-trained LLM. Thirdly, synthetic data demonstrates generalizability across categories, effectively mitigating various biases.

**Questions To Authors:**

No questions.

**Reasons To Accept:**

This paper introduces a novel approach that utilizes ChatGPT to generate synthetic training data, with the aim of enhancing the debiasing of Large Language Models (LLMs).

Experimental results indicate that the proposed approach can achieve better performance.

**Reasons To Reject:**

The proposed approach lacks technical novelty, as it relies on relatively simple yet effective prompting.

The study could benefit from more in-depth case studies and analysis to help readers better understand the advantages and disadvantages of the proposed approach.

---

> ### Author Rebuttal · Authors · 2024-05-29
>
> Dear Reviewer,
>
> Thank you for your recognition and constructive comments. We appreciate your feedback and would like to address the concerns raised.
>
> **Regarding the technical novelty**, we would like to emphasize that our contribution extends beyond simply the use of effective prompting. We present a holistic and easy-to-use pipeline that integrates knowledge about social bias specification, synthetic data generation in a specific format of [Sentence, Target, Attribute] that supports debiasing, and the associated adapter-based lightweight fine-tuning.
>
> We propose distinct strategies for synthetic data generation, each with unique strengths. Our experiments with loss-guided prompting further improve our approach’s effectiveness and applicability. This process requires an understanding of bias specification, effective adaptation of LLM capabilities, and a thoughtfully designed pipeline that combines these elements. These efforts enable both domain and non-domain experts to quickly and efficiently generate synthetic data to debias models with minimal cost.
>
> **For more in-depth analysis and case studies**, we included various levels of analysis and cases in the paper. Key experiments already included:
>
> - Reducing Bias: Effectiveness in reducing bias in BERT and four GPT models across two datasets (Figures 1 and 2) with additional analysis on a third dataset capturing different biases (Table 5).
> - Impact on Language Capabilities: Analyzing debiasing impact on model language capabilities (Tables 2-4).
> - Data Similarity Analysis: Comparing our synthetic datasets to existing datasets to address memorization and data leakage (Section 6 and Table 6).
> - Example Cases: Analysis of generated terms and sentences in our synthetic data (Figure 4, Table 1, and 7).
> - Loss-Guided Technique: Testing the hypothesis on the importance of in-domain synthetic data generation (Section 3 “Loss-Guided Prompting”).
> - Trade-Off Analysis: Evaluating the trade-off between language capabilities and debiasing performance, and performance with different training data sizes (Figures 5 and 6).
>
> We hope this extensive set of experiments provides sufficient evidence of our approach’s effectiveness. We would also appreciate any concrete suggestions for additional experiments to strengthen our results.
>
> Thank you once again for your constructive feedback! We found it to be very helpful. Feel free to let us know if there are any further concerns or if any additional adjustments are needed.

---

> > ### Comment · Reviewer_fjaf · 2024-06-05
> >
> > Thank you for your reply, I have no further questions.  I agree with Reviewer n66S that  the generated sentences might still be biased or toxic. I will remain my scores.

---

> > > ### Author Response · Authors · 2024-06-06
> > > **Additional analysis around toxicity**
> > >
> > > We would like to, once again, thank the reviewer for relating to our rebuttal and providing additional comments. Regarding the potential bias, we would like to point to the fact that the same synthetic dataset (synthetic-general) is effective at debiasing across several bias categories in different datasets, which suggests it doesn’t introduce new biases while mitigating others.
> > >
> > > However, in light of the concerns around the potential toxicity of our generations, we performed additional analysis using the unbiased [ToxicBert]([https://huggingface.co/unitary/toxic-bert) model (i.e., Detoxicty) as well as a non-neural model for sentiment analysis - [VaderSentiment]([https://github.com/cjhutto/vaderSentiment). We present these results in the table below:
> > >
> > > | Evaluation Model |    Synthetic Data|     Mean Score (SD) | Top 95% percentile |
> > > | :---- | :----  |-----------------:|----------------------: |
> > > | ToxicBERT *(toxicity)*  | General   | 0.0086 ± 0.0539 | 0.0216 |
> > > |                  | Gender   |   0.0312 ± 0.1147 | 0.1480 |
> > > |                  |  Race  | 0.0177 ± 0.0632 | 0.2943 |
> > > |                  | Religion | 0.0123 ± 0.0614 | 0.0314|
> > > |VaderSentiment  | General | 0.0081 ± 0.0404 | 0.0000 |
> > > | *(negative component)*                        | Gender   | 0.0318 ± 0.0862 | 0.2421 |
> > > |                          | Race | 0.0039 ± 0.0277 | 0.0000 |
> > > |                          | Religion | 0.0049 ± 0.0350 | 0.0000 |
> > >
> > > The Toxicity and VaderSentiment scores take the value from 0.0 to 1.0 on a continuous scale with values close to 0.0 representing very little toxicity and lack of negative sentiment respectively. We can see that the scores for our synthetic data are very low, way below 0.50 general threshold for considering a text toxic. This is even for the top 95% percentile of highest-scoring generations. This suggests that our synthetic generations are not toxic. We will be happy to add this analysis to the paper.
> > >
> > > *Methodology notes:* VaderSentiment produces individual positive, negative, and neutral sentiment component scores and a compound score. We report just the values for the negative sentiment component as these are the most relevant here. We also analyzed other dimensions provided by ToxicBert such as Identity Attack, Insult, and Threat. The scores were the highest for Toxicity (but still very low in overall terms) so we report these.
> > >
> > > We hope this additional analysis helps to alleviate concerns about toxicity.

---

### Decision · Program_Chairs · 2024-07-10

**Decision:**

Accept

**Comment:**

The paper proposed a prompting method to generate anti-stereotype sentences (500 sentences for each of the three types of biases: rase, gender, religion) as data augmentation. The synthetically generated data is then used in adapter tuning of GPT-2 and BERT models to de-bias them.

The paper presents an interesting and effective idea and a good amount of meaningful comparison experiments. I think there is enough novelty in the paper.

While the authors addressed some of the reviewer's concerns (the most effective part of the author's response was by providing the automatic toxicity analysis), the paper could be further improved by taking some of the reviewer's constructive criticism as suggestions for improvement, in particular:

(1) provide more in-depth analysis (reviewer fjaf):  In addition to the post-hoc analysis that focuses on automatic experiment results or analysis, provide some more direct analysis, including manual analysis or human evaluation, about the quality of these anti-stereotype sentences. Explain more clearly, why and where exactly your method works better than others, and where it may fail (i.e., limitations of the proposed method).

(2) update Section 2.2 to provide a more direct/comprehensive review of debiasing methods (reviewer P4Tc): include more related works and provide a discussion about the connection and difference from Attrprompt (Yu, Yue et al. 2024).

I recommend the paper for acceptance, with slight reservations on whether the authors will be willing to address the above two issues.